# Ψ-Sampler: Initial Particle Sampling for SMC-Based Inference-Time Reward Alignment in Score Models

**Taehoon Yoon**[*]   **Yunhong Min**[*]   **Kyeongmin Yeo**[*]   **Minhyuk Sung**

KAIST

{taehoon,dbsghd363,aaaaa,mhsung}@kaist.ac.kr

## Abstract

We introduce Ψ-SAMPLER, an SMC-based framework incorporating pCNL-based initial particle sampling for effective inference-time reward alignment with a score-based generative model. Inference-time reward alignment with score-based generative models has recently gained significant traction, following a broader paradigm shift from pre-training to post-training optimization. At the core of this trend is the application of Sequential Monte Carlo (SMC) to the denoising process. However, existing methods typically initialize particles from the Gaussian prior, which inadequately captures reward-relevant regions and results in reduced sampling efficiency. We demonstrate that initializing from the reward-aware posterior significantly improves alignment performance. To enable posterior sampling in high-dimensional latent spaces, we introduce the preconditioned Crank–Nicolson Langevin (pCNL) algorithm, which combines dimension-robust proposals with gradient-informed dynamics. This approach enables efficient and scalable posterior sampling and consistently improves performance across various reward alignment tasks, including layout-to-image generation, quantity-aware generation, and aesthetic-preference generation, as demonstrated in our experiments.
Project Webpage: https://psi-sampler.github.io/

## 1   Introduction

Recently, a shift in the scaling law paradigm from pre-training to post-training has opened new possibilities for achieving another leap in AI model performance, as exemplified by the unprecedented AGI score of GPT-o3 [1] and DeepSeek's "Aha moment" [2]. Breakthroughs in LLMs have also extended to score-based generative models [3–7], resulting in significant improvements in user preference alignment [8]. Similar to the autoregressive generation process in LLMs, the denoising process in score-based generative models can be interpreted as a Sequential Monte Carlo (SMC) [9–11] process with a single particle at each step. This perspective allows inference-time alignment to be applied analogously to LLMs by populating multiple particles at each step and selecting those that score highly under a given reward function [8, 12–17]. A key distinction is that score-based generative models enable direct estimation of the final output from any noisy intermediate point via Tweedie's formula [18], facilitating accurate approximation of the optimal value function [19, 8, 20] through expected reward estimation.

However, previous SMC-based approaches [12, 15, 14, 21, 22], where each SMC step is coupled with the denoising process of score-based generative models, are limited in their ability to effectively explore high-reward regions, as the influence of the reward signal diminishes over time due to vanishing diffusion coefficient. Thus, rather than relying on particle exploration during later stages, it is more critical to identify effective initial latents that are well-aligned with the reward model

---

[*]Equal contribution.

39th Conference on Neural Information Processing Systems (NeurIPS 2025).

from the outset. In this work, we address this problem and propose an MCMC-based initial particle population method that generates strong starting points for the subsequent SMC process. This direction is particularly timely given recent advances in distillation techniques for score-based generative models [23–27], now widely adopted in state-of-the-art models [28, 29]. These methods yield straighter generative trajectories and clearer Tweedie estimates [18] from early steps, enabling more effective exploration from the reward-informed initial distribution.

A straightforward baseline for generating initial particles is the Top-$K$-of-$N$ strategy: drawing multiple samples from the standard Gaussian prior and selecting those with the highest reward scores. Though effective, this naive approach offers limited improvement in subsequent SMC due to its reliance on brute-force sampling. Motivated by these limitations, we explore Markov Chain Monte Carlo (MCMC) [30–35] methods based on Langevin dynamics, which are particularly well-suited to our setting since we sample from the initial posterior distribution, whose form is known. Nevertheless, applying MCMC in our problem presents unique challenges: the exploration space is extremely high-dimensional (*e.g.*, 65,536 for FLUX [28]), posing significant challenges for conventional MCMC methods. In particular, the Metropolis–Hastings (MH) accept-reject mechanism, when used with standard Langevin-based samplers, becomes ineffective in such high-dimensional regimes, as the acceptance probability rapidly diminishes and most proposals are rejected.

Our key idea for enabling effective particle population from the initial reward-informed distribution is to leverage the Preconditioned Crank–Nicolson (pCN) algorithm [36–38], which is designed for function spaces or infinite-dimensional Hilbert spaces. When combined with the Langevin algorithm (yielding pCNL), its semi-implicit Euler formulation allows for efficient exploration in a high-dimensional space. Furthermore, when augmented with the MH correction, the acceptance rate is significantly improved compared to vanilla MALA. We therefore propose performing pCNL over the initial posterior distribution and selecting samples at uniform intervals along the resulting Markov chain. These samples are then used as initial particles for the subsequent SMC process across the denoising steps. We refer to the entire pipeline—**P**CNL-based initial particle sampling followed by **S**MC-based **I**nference-time reward alignment—as **PSI** ($\Psi$)-Sampler. To the best of our knowledge, this is the first work to apply the pCN algorithm in the context of generative modeling.

In our experiments, we evaluate three reward alignment tasks: layout-to-image generation (placing objects in designated bounding boxes within the image), quantity-aware generation (aligning the number of objects in the image with the specified count), and aesthetic-preference generation (enhancing visual appeal). We compare our $\Psi$-Sampler against the base SMC method [14] with random initial particle sampling, SMC combined with initial particle sampling via Top-$K$-of-$N$, ULA, and MALA, as well as single-particle methods [39, 40]. Across all tasks, $\Psi$-Sampler consistently achieves the best performance in terms of the given reward and generalizes well to the held-out reward, matching or surpassing existing baselines. Its improvement over the base SMC method highlights the importance of posterior-based initialization, while its outperformance over ULA and MALA further confirms the limitations of these methods in extremely high-dimensional spaces.

## 2 Related Work

### 2.1 Inference-Time Reward Alignment

Sequential Monte Carlo (SMC) [9–11] has proven effective in guiding the generation process of score-based generative models for inference-time reward alignment [22, 21, 15, 14, 12]. Prior SMC-based methods differ in their assumptions and applicability. For instance, FPS [15] and MCGdiff [22] are specifically designed for linear inverse problems and thus cannot generalize to arbitrary reward functions. SMC-Diff [21] depends on the idealized assumption that the learned reverse process exactly matches the forward noising process—an assumption that rarely holds in practice. TDS [14] and DAS [12] employ twisting and tempering strategies respectively to improve approximation accuracy while reducing the number of required particles. Despite these variations, all aforementioned SMC-based approaches share a common limitation that they initialize particles from the standard Gaussian prior, which is agnostic to the reward function. This mismatch can result in poor coverage of high-reward regions and reduced sampling efficiency.

In addition to multi-particle systems like SMC, single-particle approaches have also been explored for inference-time reward alignment [39–42]. These methods guide generation by applying reward gradients along a single sampling trajectory. However, they are inherently limited in inference-time

reward alignment, as simply increasing the number of denoising steps does not consistently lead to better sample quality. In contrast, SMC-based methods allow users to trade computational cost for improved reward alignment, making them more flexible and scalable in practice.

## 2.2 Fine-Tuning-Based Reward Alignment

Beyond inference-time methods, another line of work focuses on fine-tuning score-based generative models for reward alignment. Some approaches perform supervised fine-tuning by weighting generated samples according to their reward scores and updating the model to favor high-reward outputs [43, 44], while others frame the denoising process as a Markov Decision Process (MDPs) and apply reinforcement learning techniques such as policy gradients [45] or entropy-regularized objectives to mitigate overoptimization [46–48]. These RL-based methods are especially useful when the reward model is non-differentiable but may miss gradient signals when available. More recent methods enable direct backpropagation of reward gradients through the generative process [49, 50]. Alternatively, several works [20, 51, 52] adopt a stochastic optimal control (SOC) perspective, deriving closed-form optimal drift and initial distributions using pathwise KL objectives.

While fine-tuning-based methods are an appealing approach, they have practical limitations in that they necessitate costly retraining whenever changes are made to the reward function or the pretrained model. Further, it has been shown that fine-tuning-based methods exhibit mode-seeking behavior [12], which leads to low diversity in the generated samples.

# 3 Problem Definition & Background

## 3.1 Background: Score-Based Generative Models

Given a standard Gaussian distribution $p_1 = \mathcal{N}(\mathbf{0}, \mathbf{I})$ and data distribution $p_0$, score-based generative models are trained to estimate the score function, which is the gradient of log-density, at intermediate distributions $p_t$ along a probability path connecting $p_1$ to $p_0$.

In score-based generative models [3–7], the data generation process is typically described by a reverse-time stochastic differential equation (SDE) [4]:

$$\mathrm{d}\mathbf{x}_t = \mathbf{f}(\mathbf{x}_t, t)\mathrm{d}t + g(t)\mathrm{d}\mathbf{W}, \quad \mathbf{f}(\mathbf{x}_t, t) = \mathbf{u}(\mathbf{x}_t, t) - \frac{g(t)^2}{2}\nabla \log p_t(\mathbf{x}_t), \quad \mathbf{x}_1 \sim p_1 \qquad (1)$$

where $\mathbf{f}(\mathbf{x}_t, t)$ and $g(t)$ denote the drift and diffusion coefficients, respectively, and $\mathbf{W}$ is a $d$-dimensional standard Brownian motion. The term $\mathbf{u}(\mathbf{x_t}, \mathbf{t})$ corresponds to the velocity field in flow-based model [53, 27, 54] and also corresponds to the drift term of the probability flow ODE (PF-ODE) in diffusion models [4]. We assume that the generation process proceeds in decreasing time, $i.e.$, from $t = 1$ to $t = 0$, following the convention commonly adopted in the score-based generative modeling literature [4, 5].

The deterministic flow-based generative model can be recovered by setting the diffusion coefficient $g(t) = 0$, thereby reducing the SDE to an ODE. Note that flow-based models [53, 27], originally formulated as an ODE, can be extended to an SDE formulation that shares the same intermediate distributions $p_t$, thereby allowing stochasticity to be introduced during generation [7, 6, 13]. Moreover, the velocity field $\mathbf{u}(\mathbf{x}_t, t)$ can be readily transformed into a score function [7, 54]. For these reasons, we categorize both diffusion and flow-based models as the score-based generative models.

## 3.2 Inference-Time Reward Alignment Using Score-Based Generative Models

Inference-time reward alignment [8, 12–17] aims to generate high-reward samples $\mathbf{x}_0 \in \mathbb{R}^d$ without fine-tuning the pretrained score-based generative model. The reward associated with each sample is evaluated using a task-specific reward function $r : \mathbb{R}^d \to \mathbb{R}$, which may quantify aspects such as aesthetic quality or the degree to which a generated image satisfies user-specified conditions.

But to avoid over-optimization [46, 20, 12] with respect to the reward function, which may lead to severe distributional drift or adversarial artifacts, a regularization term is introduced to encourage the generated samples to remain close to the prior of the pre-trained generative model. This trade-off is captured by defining a target distribution $p_0^*$ that balances reward maximization with prior adherence,

formally expressed as:

$$p_0^* = \arg\max_{q} \underbrace{\mathbb{E}_{\mathbf{x}_0 \sim q}[r(\mathbf{x}_0)]}_{(a)} - \alpha \underbrace{\mathcal{D}_{\mathrm{KL}}[q\|p_0]}_{(b)}. \tag{2}$$

Here, term $(a)$ in Eq. 2 encourages the generation of high-reward samples, while term $(b)$, the KL-divergence, enforces proximity to the pre-trained model's prior distribution $p_0$. The parameter $\alpha \in \mathbb{R}_+$ controls the strength of this regularization: larger values of $\alpha$ lead to stronger adherence to the prior, typically resulting in lower reward but higher proximity to the support of the generative model.

The target distribution $p_0^*$ has a closed-form expression, given by:

$$p_0^*(\mathbf{x}_0) = \frac{1}{Z_0} p_0(\mathbf{x}_0) \exp\left(\frac{r(\mathbf{x}_0)}{\alpha}\right) \tag{3}$$

where $Z_0$ is normalizing constant. Detailed derivation using calculus of variations can be found in Kim *et al.* [13]. This reward-aware target distribution has been widely studied in the reinforcement learning literature [55–59]. Analogous ideas have also been adopted to fine-tuning score-based generative models [60, 51, 52, 45, 48–50, 46, 43, 20]. As in our case, this target distribution also serves as the objective from which one aims to sample in inference-time reward alignment task [8, 12, 13].

Since sample generation in score-based models proceeds progressively through a sequence of timesteps, it becomes important to maintain proximity with the pretrained model not just at the endpoint, but throughout the entire generative trajectory. To account for this, the original objective in Eq. 2 is extended to a trajectory-level formulation. Although there are some works [50, 46, 45] that frame this problem as entropy-regularized Markov Decision Process (MDPs), where each denoising step of score-based generative model corresponds to a policy in RL, we adopt a stochastic optimal control (SOC) perspective [20, 51, 52], which naturally aligns with the continuous-time structure of score-based generative models and yields principled expressions for both the optimal drift and the optimal initial distribution.

Building on this, the entropy-regularized SOC framework proposed by Uehara *et al.* [20] provides closed-form approximations for the optimal initial distribution, optimal control function, and optimal transition kernel that together enable sampling from the reward-aligned target distribution defined in Eq. 3 using score-based generative models.

The optimal initial distribution can be derived using the Feynman–Kac formula and approximated via Tweedie's formula [18] as:

$$\tilde{p}_1^*(\mathbf{x}_1) := \frac{1}{Z_1} p_1(\mathbf{x}_1) \exp\left(\frac{r(\mathbf{x}_{0|1})}{\alpha}\right) \tag{4}$$

where $\mathbf{x}_{0|t} := \mathbb{E}_{\mathbf{x}_0 \sim p_{0|t}}[\mathbf{x}_0]$ denotes Tweedie's formula [18], representing the conditional expectation under $p_{0|t} := p(\mathbf{x}_0|\mathbf{x}_t)$. Under the same approximation, the transition kernel satisfying the optimality condition is approximated by:

$$\tilde{p}_\theta^*(\mathbf{x}_{t-\Delta t}|\mathbf{x}_t) = \frac{\exp(r(\mathbf{x}_{0|t-\Delta t})/\alpha)}{\exp(r(\mathbf{x}_{0|t})/\alpha)} p_\theta(\mathbf{x}_{t-\Delta t}|\mathbf{x}_t). \tag{5}$$

where $p_\theta(\mathbf{x}_{t-\Delta t}|\mathbf{x}_t)$ is a transition kernel of the pretrained score-based generative model.

Further details on the SOC framework and its theoretical foundations in the context of reward alignment are provided in the Appendix A.

### 3.3 Sequential Monte Carlo (SMC) with Denoising Process

For reward-alignment tasks, recent works [12, 15, 14, 21, 22] have demonstrated that Sequential Monte Carlo (SMC) can efficiently generate samples from the target distribution in Eq. 3. When applied to score-based generative models, the denoising process is coupled with the sequential structure of SMC. Specifically, several prior works [8, 12, 14] adopt Eq. 5 as the intermediate target transition kernel for sampling from Eq. 3.

In general, SMC methods [9–11] are a class of algorithms for sampling from sequences of probability distributions. Starting from $K$ particles sampled $i.i.d.$ from the initial distribution, SMC approximates a target distribution by maintaining a population of $K$ weighted particles, which are repeatedly updated through a sequence of propagation, reweighting, and resampling steps. The weights are

updated over time according to the following rule:

$$w_{t-\Delta t}^{(i)} = \frac{p_{\mathrm{tar}}(\mathbf{x}_{t-\Delta t}|\mathbf{x}_t)}{q(\mathbf{x}_{t-\Delta t}|\mathbf{x}_t)} w_t^{(i)} \tag{6}$$

where $p_{\mathrm{tar}}$ is an intermediate target kernel we want to sample from, and $q(\mathbf{x}_{t-\Delta t}|\mathbf{x}_t)$ is a proposal kernel used during propagation. As the number of particles $K$ increases, the approximation improves due to the asymptotic consistency of the SMC framework [61, 62].

Following [8, 12, 14], which derives both the intermediate target transition kernel and the associated proposal for reward-guided SMC, we compute the weight at each time step as:

$$w_{t-\Delta t}^{(i)} = \frac{\exp(r(\mathbf{x}_{0|t-\Delta t})/\alpha)p_\theta(\mathbf{x}_{t-\Delta t}|\mathbf{x}_t)}{\exp(r(\mathbf{x}_{0|t})/\alpha)q(\mathbf{x}_{t-\Delta t}|\mathbf{x}_t)} w_t^{(i)}, \tag{7}$$

where $p_{\mathrm{tar}}$ is set as Eq. 5. The proposal distribution $q(\mathbf{x}_{t-\Delta t}|\mathbf{x}_t)$ is obtained by discretizing the reverse-time SDE with an optimal control. This yields the following proposal with the Tweedie's formula [18]:

$$q(\mathbf{x}_{t-\Delta t}|\mathbf{x}_t) = \mathcal{N}(\mathbf{x}_t - \mathbf{f}(\mathbf{x}_t, t)\Delta t + g^2(t)\nabla\frac{r(\mathbf{x}_{0|t})}{\alpha}\Delta t, \ g(t)^2\Delta t\mathbf{I}). \tag{8}$$

Details on SMC and its connection to reward-guided sampling are provided in the Appendix B.

### 3.4 Limitations of Previous SMC-Based Reward Alignment Methods

While prior work [12, 15, 14, 21, 22] has demonstrated the effectiveness of SMC in inference-time reward alignment, these approaches typically rely on sampling initial particles from the standard Gaussian prior. We argue that sampling particles directly from the posterior in Eq. 4, rather than the prior, is essential for better high-reward region coverage and efficiency in SMC. First, the effectiveness of the SMC proposal distribution Eq. 8 diminishes over time making it increasingly difficult to guide particles toward high-reward regions in later steps. As the diffusion coefficient $g(t)^2 \to 0$ as $t \to 0$, it weakens the influence of the reward signal $\nabla r(\mathbf{x}_{0|t})$, since it is scaled by $g^2(t)$ in the proposal. Second, the initial position of particles becomes particularly critical when the reward function is highly non-convex and multi-modal. While the denoising process may, in principle, help particles escape local modes and explore better regions, this becomes increasingly difficult over time, not only due to the vanishing diffusion coefficient, but also because the intermediate distribution becomes less perturbed and more sharply concentrated, reducing connectivity between modes [63]. In contrast, at early time steps (*e.g.*, $t = 1$), the posterior distribution is more diffuse and better connected across modes, enabling more effective exploration. Furthermore, recent score-based generative models distilled for trajectory straightening have made the approximation of the optimal initial distribution in Eq. 4 sufficiently precise. These observations jointly motivate allocating computational effort to obtaining high-quality initial particles that are better aligned with the reward signal.

## 4 $\Psi$-Sampler: pCNL-Based Initial Particle Sampling

In this work, we propose $\Psi$-Sampler, a framework that combines efficient initial particle samping with SMC-based inference-time reward alignment for score-based generative models. The initial particles are sampled using the Preconditioned Crank–Nicolson Langevin (pCNL) algorithm, hence the name **P**CNL-based initial particle sampling followed by **S**MC-based **I**nference-time reward alignment. The key idea is to allocate computational effort to the initial particle selection by sampling directly from the posterior distribution defined in Eq. 4. This reward-informed initialization ensures that the particle set is better aligned with the target distribution from the outset, resulting in improved sampling efficiency and estimation accuracy in the subsequent SMC process.

While the unnormalized density of the posterior distribution in Eq. 4 has an analytical form, drawing exact samples from it remains challenging. A practical workaround is to approximate posterior sampling via a **Top-$K$-of-$N$** strategy: generate $N$ samples from the prior, and retain the top $K$ highest-scoring samples as initial particles. This variant of Best-of-$N$ [64–66] resembles rejection sampling and serves as a crude approximation to posterior sampling [67, 68]. We find that even this simple selection-based approximation leads to meaningful improvements. But considering that sampling space is high-dimensional, one can adopt Markov Chain Monte Carlo (MCMC) [30–35] which is known to be effective at sampling from high-dimensional space. In what follows, we briefly introduce Langevin-based MCMC algorithms that we adopt for posterior sampling.

## 4.1 Background: Langevin-Based Markov Chain Monte Carlo Methods

Langevin-based MCMC refers to a class of samplers that generate proposals by discretizing the Langevin dynamics, represented as stochastic differential equation (SDE),

$$d\mathbf{x} = \frac{1}{2}\nabla \log p_{\text{tar}}(\mathbf{x})\, dt + d\mathbf{W}, \tag{9}$$

whose stationary distribution is the target density $p_{\text{tar}}$. A single Euler–Maruyama discretization of the Langevin dynamics with step size $\epsilon > 0$ produces the proposal

$$\mathbf{x}' = \mathbf{x} + \frac{\epsilon}{2}\nabla \log p_{\text{tar}}(\mathbf{x}) + \sqrt{\epsilon}\,\mathbf{z}, \qquad \mathbf{z} \sim \mathcal{N}(\mathbf{0}, \mathbf{I}). \tag{10}$$

Accepting every proposal yields the **Unadjusted Langevin Algorithm (ULA)** [31]. As a result, the Markov chain induced by ULA converges to a biased distribution whose discrepancy arises from the discretization error. In particular, since ULA does not include a correction mechanism, it does not guarantee convergence to the target distribution $p_{\text{tar}}$. **Metropolis-Adjusted Langevin Algorithm (MALA)** [31, 32] combines the Langevin proposal Eq. 10 with the Metropolis–Hastings (MH) [69, 30] correction, a general accept–reject mechanism that eliminates discretization bias. Given the current state $\mathbf{x}$ and a proposal $\mathbf{x}' \sim q(\mathbf{x}'|\mathbf{x})$, the move is accepted with probability:

$$a_{\mathbf{M}}(\mathbf{x}, \mathbf{x}') = \min\left(1, \frac{p_{\text{tar}}(\mathbf{x}')\, q(\mathbf{x}|\mathbf{x}')}{p_{\text{tar}}(\mathbf{x})\, q(\mathbf{x}'|\mathbf{x})}\right). \tag{11}$$

This rule enforces detailed balance, so $p_{\text{tar}}$ is an invariant distribution of the resulting Markov chain. While MALA is commonly used in practice due to its simplicity and gradient-based efficiency, it becomes increasingly inefficient in extremely high-dimensional settings, as is typical in image generative models (*e.g.*, 65,536 for FLUX [28]). With a fixed step size, its acceptance probability degenerates as $d \to \infty$. Theoretically, to maintain a reasonable acceptance rate, the step size must shrink with dimension, typically at the optimal rate of $O(d^{-1/3})$ [70], which leads to extremely slow mixing and inefficient exploration in extremely high-dimension space.

## 4.2 Preconditioned Crank–Nicolson Langevin (pCNL) Algorithm

To address high-dimensional sampling challenges (Sec. 4.1), infinite-dimensional MCMC methods [36–38] were developed, particularly for PDE-constrained Bayesian inverse problems. These methods remain well-posed even when dimensionality increases. Among them, the preconditioned Crank–Nicolson (pCN) algorithm offers a simple, dimension-robust alternative to Random Walk Metropolis (RWM), though it fails to leverage the potential function, limiting its efficiency.

To overcome this limitation, the preconditioned Crank–Nicolson Langevin (pCNL) algorithm has been proposed [36, 37], which augments the dimension-robustness of pCN with the gradient-informed dynamics of Langevin methods (Eq. 9), thereby improving sampling efficiency in high-dimensional settings. The pCNL algorithm employs a semi-implicit Euler (Crank–Nicolson-type) discretization of Langevin dynamics as follows:

$$\mathbf{x}' = \mathbf{x} + \frac{\epsilon}{2}\left(-\frac{\mathbf{x} + \mathbf{x}'}{2} + \nabla\frac{r(\mathbf{x}_{0|1})}{\alpha}\right) + \sqrt{\epsilon}\,\mathbf{z}, \qquad \mathbf{z} \sim \mathcal{N}(\mathbf{0}, \mathbf{I}). \tag{12}$$

assuming prior is $\mathcal{N}(\mathbf{0}, \mathbf{I})$ as in our case. This Crank–Nicolson update admits an explicit closed-form solution, and hence retains the dimension-robustness of pCN, only when the drift induced by the prior is linear, as with a standard Gaussian prior. Therefore, in our setting, it is applicable only at $t = 1$, making it a particularly useful method that aligns with our proposal to sample particles from the posterior distribution Eq. 4, where the prior is the standard Gaussian. With $\rho = (1 - \epsilon/4)/(1 + \epsilon/4)$, we can rewrite above equation as:

$$\mathbf{x}' = \rho\mathbf{x} + \sqrt{1 - \rho^2}\left(\mathbf{z} + \frac{\sqrt{\epsilon}}{2}\nabla\frac{r(\mathbf{x}_{0|1})}{\alpha}\right), \qquad \mathbf{z} \sim \mathcal{N}(\mathbf{0}, \mathbf{I}). \tag{13}$$

Note that pCNL also adopts MH correction in Eq. 11 to guarantee convergence to the correct target distribution. The pCN algorithm maintains a well-defined, non-zero acceptance probability even in the infinite-dimensional limit, allowing the use of fixed step sizes regardless of the dimension $d$ [36, 37]. This property stems from its prior-preserving proposal, which ensures that the Gaussian reference measure is invariant under the proposal mechanism. This robustness carries over to pCNL, whose proposal inherits pCN's ability to handle Gaussian priors in a dimension-independent manner. We include the detailed acceptance probability formulas for MALA and pCNL in the Appendix C.

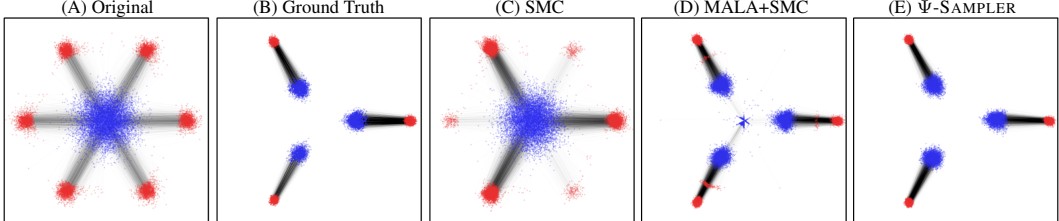

Figure 1: **Toy sampling–method comparison.** Each panel visualizes both the initial samples (blue) and their corresponding clean data samples (red). From left to right: (A) samples from the original score-based generative model; (B) the target distribution defined by Eq. 3; (C) results from SMC; (D) results from MALA+SMC; and (E) results from our proposed $\Psi$-Sampler.

## 4.3 Initial Particle Sampling

To sample initial particles using MCMC for the subsequent SMC process, we follow standard practices to ensure effective mixing and reduce sample autocorrelation. Specifically, we discard the initial portion of each chain as burn-in [71] and apply thinning by subsampling at fixed intervals to mitigate high correlation between successive samples. A constant step size is used across iterations. Although adaptive step size schemes may improve convergence, we opt for a fixed-step approach for simplicity. Once the initial particles are sampled, we apply the existing SMC-based method [14, 8].

**Comparison of SMC Initialization in a Toy Experiment.**    In Fig. 1, we present a 2D toy experiment comparing SMC performance when initializing particles from the prior versus the posterior. We train a simple few-step score-based generative model on a synthetic dataset where the clean data distribution $p_0$ is a 6-mode Gaussian Mixture Model (GMM), shown as red dots in Fig.1 (A). The prior distribution is shown in blue, and the gray lines depict sampling trajectories during generation. We define a reward function that assigns high scores to samples from only a subset of the GMM modes, yielding a target distribution at $t = 0$ (Eq. 3), as illustrated in Fig. 1 (B) (red dots). The corresponding optimal initial distribution—the posterior at $t = 1$ (Eq. 4)—is shown as blue dots in Fig. 1 (B). We compare (C) standard SMC with prior sampled particles, (D) SMC with posterior samples from MALA, and (E) our $\Psi$-Sampler. All settings use the same total number of function evaluations (NFE). Prior-based SMC (C) uses 100 NFE; MALA+SMC and $\Psi$-Sampler allocate 50 NFE for MCMC and use fewer particles for SMC. While MALA-based initialization method (D) significantly improves alignment with the target distribution (red dots in (B)) over prior-based method (C), some modes remain underrepresented. In contrast, $\Psi$-Sampler (E) provides tighter alignment with the target distribution and the posterior distribution, illustrating its effectiveness in sampling high-quality samples. Full experimental details are provided in Appendix E.

# 5 Experiments

## 5.1 Experiment Setup

We validate our approach across three applications: layout-to-image generation, quantity-aware generation, and aesthetic-preference image generation. In our experiments, the *held-out* reward refers to an evaluation metric that is not accessible during generation and is used solely to assess the generalization of the method. Full details for each application are provided in Appendix D.

For the layout-to-image generation task, where the goal is to place user-specified objects within designated bounding boxes, we use predicted bounding box information from a detection model [72] and define the reward as the mean Intersection-over-Union (mIoU) between the predicted and target bounding boxes. For the quantity-aware image generation task, which involves generating a user-specified object in a specified quantity, we use the predicted count from a counting model [73] and define the reward as the negative smooth L1 loss between the predicted and target counts. In both tasks, we include evaluations using held-out reward models to assess generalization. Specifically, for layout-to-image generation, we report mIoU evaluated with a different detection model [74] (held-out reward model); for quantity-aware image generation, we report mean absolute error (MAE) and counting accuracy using an alternative counting model [75] (held-out reward model). For aesthetic-preference image generation task, which aims to produce visually appealing images, we use an aesthetic score prediction model [76] as the reward model and use its predicted score as the reward. Across all applications, we further evaluate the generated images using ImageReward [77]

| Tasks | Metrics | Single Particle | | SMC-Based Methods | | | | | |
| | | | | Sampling from Prior | | Sampling from Posterior | | | |
| | | DPS [39] | FreeDoM [40] | TDS [14] | DAS [12] | Top-$K$-of-$N$ | ULA | MALA | $\Psi$-Sampler |
|---|---|---|---|---|---|---|---|---|---|
| **Layout to Image** | GroundingDINO[†] [72] ↑ | 0.166 | 0.177 | 0.417 | 0.363 | 0.425 | 0.370 | 0.401 | **0.467** |
| | mIoU [74] ↑ | 0.215 | 0.229 | 0.402 | 0.342 | 0.427 | 0.374 | 0.401 | **0.471** |
| | ImageReward [77] ↑ | 0.705 | 0.713 | 0.962 | 0.938 | 0.957 | 0.838 | 0.965 | **1.035** |
| | VQA [78] ↑ | 0.684 | 0.650 | 0.794 | 0.784 | **0.855** | 0.783 | 0.789 | 0.810 |
| **Quantity Aware** | T2I-Count[†] [73] ↓ | 14.187 | 15.214 | 1.804 | 1.151 | 1.077 | 3.035 | 1.601 | **0.850** |
| | MAE [75] ↓ | 15.7 | 15.675 | 5.3 | 4.175 | 3.675 | 4.825 | 3.575 | **2.925** |
| | Acc (%) [75] ↑ | 0.0 | 0.0 | 27.5 | 15.0 | 12.5 | 22.5 | 25.0 | **32.5** |
| | ImageReward [77] ↑ | 0.746 | 0.665 | 0.656 | 0.507 | 0.752 | 0.743 | 0.742 | **0.796** |
| | VQA [78] ↑ | 0.957 | 0.953 | 0.943 | 0.907 | **0.960** | 0.943 | 0.941 | 0.951 |
| **Aesthetic Preference** | Aesthetic[†] [76] ↑ | 6.139 | 6.310 | 6.853 | 6.935 | 6.879 | 6.869 | 6.909 | **7.012** |
| | ImageReward [77] ↑ | 1.116 | 1.132 | 1.135 | 1.166 | 1.133 | 1.100 | 1.155 | **1.171** |
| | VQA [78] ↑ | 0.968 | 0.959 | **0.970** | **0.970** | 0.961 | 0.961 | 0.952 | 0.963 |

Table 1: Quantitative comparison of $\Psi$-Sampler and baselines across three task domains. **Bold** indicates the best performance, while underline denotes the second-best result for each metric. Metrics marked with [†] are used as seen reward during reward-guided sampling, where others are held-out reward. Higher values indicate better performance (↑), unless otherwise noted (↓).

and VQAScore [78], which assess overall image quality and text-image alignment. The baselines and our methods are categorized into three groups:

- **Single-Particle**: DPS [39] and FreeDoM [40] are methods not based on SMC but instead use a single particle trajectory and perform gradient ascent. They are limited in scaling up the search space due to the use of a single particle.

- **SMC & Initial Particles from Prior**: TDS [14] is the SMC-based method we take as the base for our methods. DAS [12] is a variant introducing tempering strategy.

- **SMC & Initial Particles from Posterior**: We evaluate four posterior-based initialization strategies: Top-$K$-of-$N$, ULA, MALA, and $\Psi$-Sampler. ULA and MALA use a small step size (0.05) to ensure non-zero acceptance, while $\Psi$-Sampler employs a larger step size (0.5) for improved performance (See Sec. 5.4).

We use 25 denoising steps for SMC-based methods and 50 for single-particle methods to compensate for their limited exploration. For SMC-based methods, we match the total number of function evaluations (NFE) across all methods, allocating half of the budget to initial particle sampling for posterior-based methods. We use FLUX [28] as the pretrained score-based generative model. Full experimental details are provided in Appendix D.

## 5.2 Quantitative Results

We present quantitative results in Tab.1. Across all tasks, $\Psi$-Sampler consistently achieves the best performance on the given reward and strong generalization to held-out rewards. For SMC-based methods, sampling particles from the posterior distribution yields significant improvements over those that sample directly from the prior, highlighting the importance of posterior-informed initialization. This improvement is particularly notable in complex tasks where high-reward outputs are rare, such as layout-to-image generation and quantity-aware generation. For example, in quantity-aware generation, negative smooth L1 loss improves from 1.804 with TDS (base SMC) to 1.077 with Top-$K$-of-$N$ and further to 0.850 with our $\Psi$-Sampler. Similarly, for layout-to-image generation, mIoU increases from 0.417 (TDS) to 0.425 with Top-$K$-of-$N$ and 0.467 with $\Psi$-Sampler. In contrast, initializing with ULA or MALA yields only marginal gains or even degraded performance, due to the lack of Metropolis-Hastings correction in ULA and the limited exploration capacity of MALA in high-dimensional spaces. Single-particle methods consistently underperform compared to SMC-based methods.

**Ablation Study.** We conduct an ablation study that examines how performance varies under different allocations of a fixed total NFE between the initial particle sampling stage (via Top-$K$-of-$K$ or MCMC) and the subsequent SMC stage; full results and analysis are provided in Appendix G.

**Additional Results Conducted with Other Score-Based Generative Models.** We additionally provide quantitative and qualitative results on all three applications using another score-based generative model, SANA-Sprint [29] in Appendix H.

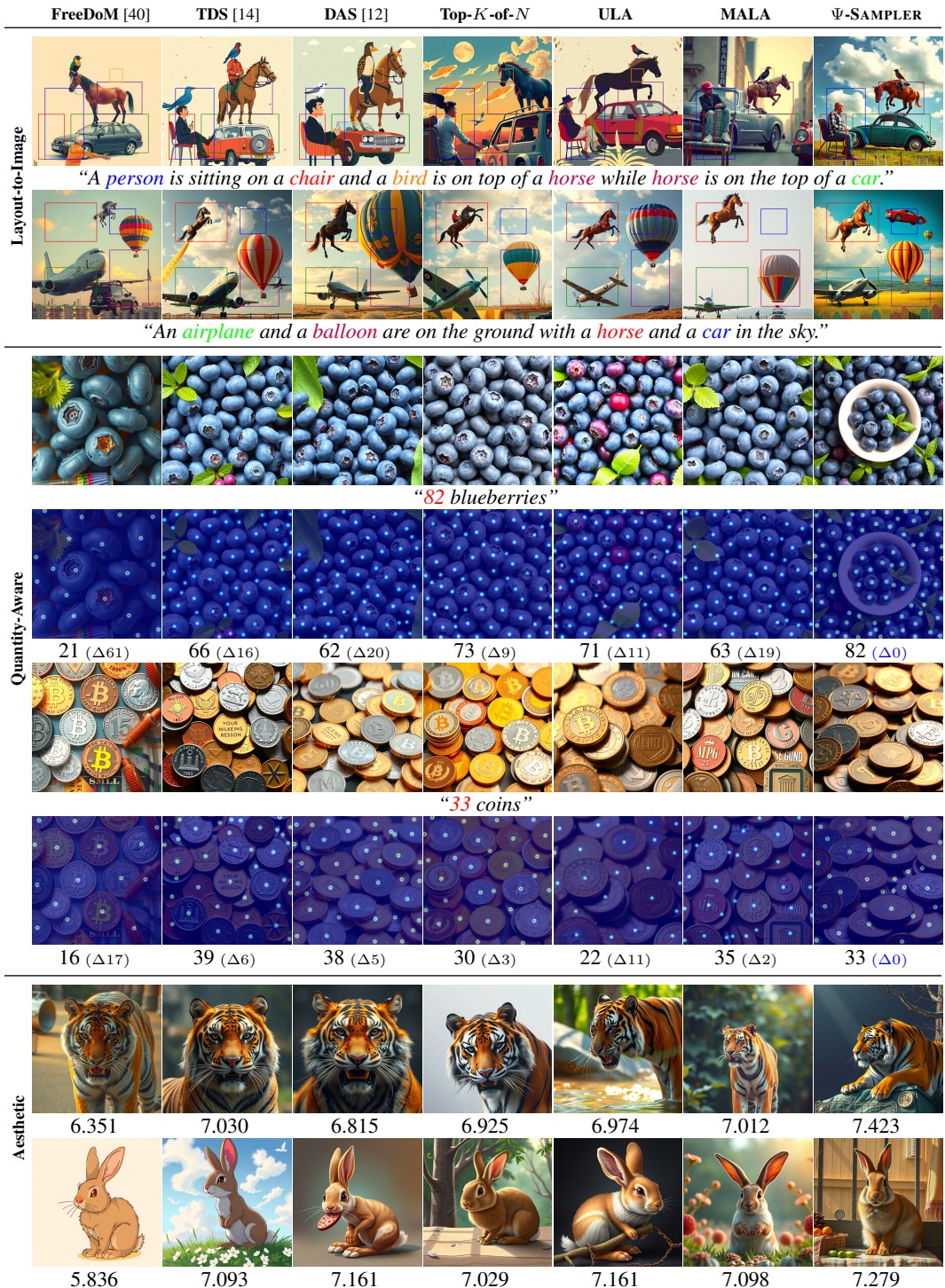

Figure 2: Qualitative results for each application demonstrate that Ψ-SAMPLER consistently generates images aligned with the given conditions. Detailed analysis of each case is provided in Sec. 5.3.

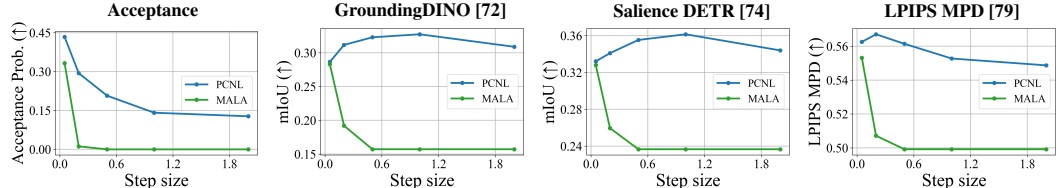

Figure 3: Performance comparison of MALA and pCNL across different evaluation metrics with varying step sizes. Conducted on *layout-to-image generation* application.

## 5.3 Qualitative Results

We additionally present qualitative results for each application in Fig. 2. For the layout-to-image generation task, we display the input bounding box locations alongside the corresponding phrases from the text prompt, using matching colors for each phrase and its associated bounding box. In the quantity-aware image generation task, we overlay the predicted object centroids—obtained from a held-out counting model [75]—to facilitate visual comparison. Below each image, we display the predicted count along with the absolute difference from the target quantity, formatted as $(\Delta \cdot)$. The best-performing case is highlighted in blue. For the aesthetic preference task, we display the generated images alongside their predicted aesthetic scores. The first row corresponds to the prompt *"Tiger"*, and the second to *"Rabbit"*. As shown, $\Psi$-SAMPLER produces high-quality results across all applications, matching the trends observed in the quantitative evaluations.

From the first and second rows of Fig. 2, we observe that baseline methods often fail to place objects correctly within the specified bounding boxes or generate them in entirely wrong locations. For instance, in the first row, most baselines fail to position the bird accurately, and in the second row, none correctly place the car. For quantity-aware generation, the fourth row shows the counted results corresponding to the third row. While $\Psi$-SAMPLER successfully generates the target number of blueberries in an image, the baselines exhibit large errors—Top-$K$-of-$N$ comes closest but still misses some. In rows 5 and 6, only $\Psi$-SAMPLER correctly generates the target number of coins. In the aesthetic preference task, although all methods produce realistic images, $\Psi$-SAMPLER generates the most visually appealing image with the highest aesthetic score. Additional qualitative examples are provided in the Appendix I.

## 5.4 Evaluation of Initial Particles

In Fig.4, we compare MALA and pCNL on the layout-to-image generation task across varying step sizes using four metrics: acceptance probability, reward (mIoU via GroundingDINO [72]), held-out reward (Salience DETR [74]), and sample diversity (LPIPS MPD [79]). All metrics are directly computed from the Tweedie estimates [18] of MCMC samples, before the SMC stage. As the step size increases, MALA's acceptance probability rapidly drops to near-zero, while pCNL maintains stable acceptance probability. Larger step sizes generally improve reward scores, with performance tapering off at excessively large steps. Held-out reward trends mirror this pattern, suggesting that the improvements stem from genuinely higher-quality samples rather than reward overfitting [60, 20]. Although LPIPS MPD slightly declines with increasing step size due to reduced acceptance, pCNL at step size 2.0 maintains diversity on par with MALA at 0.05. Additional results for other tasks are included in the Appendix F.

## 6 Conclusion and Limitation

We present a novel approach for inference-time reward alignment in score-based generative models by initializing SMC particles from the reward-aware posterior distribution. To address the challenge of high-dimensional sampling, we leverage the preconditioned Crank–Nicolson Langevin (pCNL) algorithm. Our method consistently outperforms existing baselines across tasks and reward models, demonstrating the effectiveness of posterior-guided initialization in enhancing sample quality under fixed compute budgets.

**Limitations and Societal Impact.** A limitation of our approach is that it assumes access to differentiable reward models and depends on accurate Tweedie approximations at early denoising steps. Also, while our method improves fine-grained control in generative modeling, it may also be misused to produce misleading or harmful content, such as hyper-realistic fake imagery. These risks highlights the importance of responsible development and deployment practices, including transparency, content verification, and appropriate use guidelines.

# 7 Acknowledgements

This work was supported by the NRF of Korea (RS-2023-00209723); IITP grants (RS-2019-II190075, RS-2022-II220594, RS-2023-00227592, RS-2024-00399817, RS-2025-25441313, RS-2025-25443318, RS-2025-02653113); and the Technology Innovation Program (RS-2025-02317326), all funded by the Korean government (MSIT and MOTIE), as well as by the DRB-KAIST SketchThe-Future Research Center.

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

# Appendix

## A    Reward Alignment with Stochastic Optimal Control

In the reward alignment task for continuous-time generative models [20, 52], which our method builds upon, Uehara *et al.* [20] introduce both an additional drift term $\psi$ (often referred to as a control vector field) and a modified initial distribution $\bar{p}_1$. Then the goal is to find $\psi$ and $\bar{p}_1$ such that the resulting final distribution at time $t = 0$ matches the target distribution $p_0^*$ defined in Eq. 3. Accordingly, the original reverse-time SDE used for generation (Eq. 1) is replaced by a controlled SDE:

$$\mathrm{d}\mathbf{x}_t = (\mathbf{f}(\mathbf{x}_t, t) - \psi(\mathbf{x}_t, t))\,\mathrm{d}t + g(t)\mathrm{d}\mathbf{W}, \qquad \mathbf{x}_1 \sim \bar{p}_1. \tag{14}$$

This entropy-regularized stochastic optimal control framework adopts a pathwise optimization that integrates KL divergence penalties over trajectories and thus optimization formulation of Eq. 2 changes accordingly:

$$\psi^*, p_1^* = \arg\max_{\psi, \bar{p}_1} \mathbb{E}_{\mathbb{P}^{\psi, \bar{p}_1}} \left[ r(\mathbf{x}_0) \right] - \alpha \mathcal{D}_{\mathrm{KL}} \left[ \mathbb{P}^{\psi, \bar{p}_1} \| \mathbb{P}^{\mathrm{data}} \right], \tag{15}$$

where $\mathbb{P}^{\psi, \bar{p}_1}$ is a measure over trajectories induced by the controlled SDE in Eq. 14 and $\mathbb{P}^{\mathrm{data}}$ is a measure over trajectories induced by the pre-trained SDE in Eq. 1.

KL-divergence term in Eq. 15 can be expressed as the sum of affine control cost $\frac{\|\psi(\mathbf{x}_t, t)\|^2}{g^2(t)}$ and log Radon–Nikodym derivative at $t = 1$, *i.e.*, $\log \frac{\bar{p}_1(\mathbf{x}_1)}{p_1(\mathbf{x}_1)}$:

$$\psi^*, p_1^* = \arg\max_{\psi, \bar{p}_1} \mathbb{E}_{\mathbb{P}^{\psi, \bar{p}_1}} \left[ r(\mathbf{x}_0) \right] - \alpha \mathbb{E}_{\mathbb{P}^{\psi, \bar{p}_1}} \left[ \frac{1}{2} \int_{t=0}^1 \frac{\|\psi(\mathbf{x}_t, t)\|^2}{g^2(t)} \mathrm{d}t + \log \frac{\bar{p}_1(\mathbf{x}_1)}{p_1(\mathbf{x}_1)} \right], \tag{16}$$

which can be proved [20, 52] using Girsanov theorem and martingale property of Itô integral.

The optimal control $\psi^*$ and the optimal initial distribution $p_1^*$ can be derived by introducing the optimal value function, defined as:

$$V_t^*(\mathbf{x}_t) = \max_{\psi} \mathbb{E}_{\mathbb{P}^{\psi}} \left[ r(\mathbf{x}_0) - \frac{\alpha}{2} \int_{s=0}^t \frac{\|\psi(\mathbf{x}_s, s)\|^2}{g^2(s)} \mathrm{d}s \middle| \mathbf{x}_t \right]. \tag{17}$$

where the expectation is taken over trajectories induced by the controlled SDE in Eq. 14 with current $\mathbf{x}_t$ is given.

From the optimal value function at $t = 1$, we can derive explicit formulation of the optimal initial distribution $p_1^*$ in terms of $V_1^*(\mathbf{x}_t)$ by plugging the definition of optimal value function at $t = 1$ (Eq. 17) into Eq. 16:

$$p_1^* = \arg\max_{\bar{p}_1} \mathbb{E}_{\bar{p}_1} \left[ V_1^*(\mathbf{x}_1) \right] - \alpha \mathcal{D}_{\mathrm{KL}} \left[ \bar{p}_1 \| p_1 \right]. \tag{18}$$

Solving this yields the following closed-form expression for the optimal initial distribution (derivable via calculus of variations [13]), similarly to Eq. 3:

$$p_1^*(\mathbf{x}_1) = \frac{1}{Z_1} p_1(\mathbf{x}_1) \exp \left( \frac{V_1^*(\mathbf{x}_1)}{\alpha} \right). \tag{19}$$

The optimal control $\psi^*$ can be obtained from the Hamilton–Jacobi–Bellman (HJB) equation and is expressed in terms of the gradient of the optimal value function:

$$\psi^*(\mathbf{x}_t, t) = g^2(t) \nabla \frac{V_t^*(\mathbf{x}_t)}{\alpha}. \tag{20}$$

Moreover, the optimal value function itself admits an interpretable closed-form expression via the Feynman–Kac formula:

$$V_t^*(\mathbf{x}_t) = \alpha \log \mathbb{E}_{\mathbb{P}^{\mathrm{data}}} \left[ \exp \left( \frac{r(\mathbf{x}_0)}{\alpha} \right) \middle| \mathbf{x}_t \right]. \tag{21}$$

Importantly, Uehara *et al.* [20] further proved that the marginal distribution $p_t^*(\mathbf{x}_t)$ induced by the controlled SDE (with optimal control $\psi^*$ and optimal initial distribution $p_1^*$) is given by:

$$p_t^*(\mathbf{x}_t) = \frac{1}{Z_t} p_t(\mathbf{x}_t) \exp\left(\frac{V_t^*(\mathbf{x}_t)}{\alpha}\right), \tag{22}$$

where $p_t(\mathbf{x}_t)$ is the marginal distribution of the pretrained score-based generative model at time $t$. Similarly, the optimal transition kernel under the controlled dynamics is:

$$p_\theta^*(\mathbf{x}_{t-\Delta t}|\mathbf{x}_t) = \frac{\exp(V_{t-\Delta t}^*(\mathbf{x}_{t-\Delta t})/\alpha)}{\exp(V_t^*(\mathbf{x}_t)/\alpha)} p_\theta(\mathbf{x}_{t-\Delta t}|\mathbf{x}_t), \tag{23}$$

where $p_\theta(\mathbf{x}_{t-\Delta t}|\mathbf{x}_t)$ denotes the transition kernel of the pretrained model, *i.e.*, corresponding to the discretization of the reverse-time SDE defined in Eq. 1. For detail derivation, see Theorem 1 and Lemma 3 in Uehara *et al.* [20]. Notably, Eq. 22 implies that by following the controlled dynamics defined by Eq. 14, initialized with the optimal distribution $p_1^*$ and guided by the optimal control $\psi^*$, the resulting distribution at time $t = 0$ will match the target distribution $p_0^*$ defined in Eq. 3.

Note that the optimal control, optimal initial distribution, and optimal transition kernel are all expressed in terms of the optimal value function. However, despite their interpretable forms, these expressions are not directly computable in practice due to the intractability of the posterior $p(\mathbf{x}_0|\mathbf{x}_t)$. This motivates the use of approximation techniques, most notably Tweedie's formula [18], which is widely adopted in the literature [39, 42, 14, 12] to make such expressions tractable. Under this approximation, the posterior is approximated by a Dirac-delta distribution centered at the posterior mean denoted by $\mathbf{x}_{0|t} := \mathbb{E}_{\mathbf{x}_0 \sim p_{0|t}}[\mathbf{x}_0]$, representing the conditional expectation under $p_{0|t} := p(\mathbf{x}_0|\mathbf{x}_t)$. Consequently, the optimal value function simplifies to:

$$V_t^*(\mathbf{x}_t) = \alpha \log \int \exp\left(\frac{r(\mathbf{x}_0)}{\alpha}\right) p(\mathbf{x}_0|\mathbf{x}_t)\mathrm{d}\mathbf{x}_0 \simeq \alpha \log \int \exp\left(\frac{r(\mathbf{x}_0)}{\alpha}\right) \delta(\mathbf{x}_0 - \mathbf{x}_{0|t})\mathrm{d}\mathbf{x}_0 = r(\mathbf{x}_{0|t}), \tag{24}$$

where $\mathbf{x}_{0|t}$ is a deterministic function of $\mathbf{x}_t$. Using this approximation, we have following approximations for the optimal initial distribution $\tilde{p}_1^*$, the optimal control $\tilde{\psi}^*$, and the optimal transition kernel $\tilde{p}_\theta^*$, which are used throughout the paper:

$$\tilde{p}_1^*(\mathbf{x}_1) := \frac{1}{Z_1} p_1(\mathbf{x}_1) \exp\left(\frac{r(\mathbf{x}_{0|1})}{\alpha}\right) \tag{25}$$

$$\tilde{\psi}^*(\mathbf{x}_t) = g^2(t)\nabla \frac{r(\mathbf{x}_{0|t})}{\alpha} \tag{26}$$

$$\tilde{p}_\theta^*(\mathbf{x}_{t-\Delta t}|\mathbf{x}_t) = \frac{\exp(r(\mathbf{x}_{0|t-\Delta t})/\alpha)}{\exp(r(\mathbf{x}_{0|t})/\alpha)} p_\theta(\mathbf{x}_{t-\Delta t}|\mathbf{x}_t). \tag{27}$$

It is worth noting that sampling from the optimal initial distribution is essential to theoretically guarantee convergence to the target distribution Eq. 3. Simply following the optimal control alone does not suffice and can in fact bias away from the target, a phenomenon known as the *value function bias* problem [52]. To the best of our knowledge, this is the first work to explicitly address this problem in the context of inference-time reward-alignment with score-based generative models.

## B   Sequential Monte Carlo and Reward-Guided Sampling

Sequential Monte Carlo (SMC) methods [9–11], also known as particle filter, are a class of algorithms for sampling from sequences of probability distributions. Beginning with $K$ particles drawn independently from an initial distribution, SMC maintains a weighted particle population, $\{\mathbf{x}_t^{(i)}\}_{i=1}^K$, and iteratively updates it through propagation, reweighting, and resampling steps to approximate the target distribution. During propagation, particles are moved using a proposal distribution; in the reweighting step, their importance weights are adjusted to reflect the discrepancy between the target and proposal distributions; and resampling preferentially retains high-weight particles while

eliminating low-weight ones (this is performed only conditionally, see below). The weights are updated over time according to the following rule:

$$w_{t-\Delta t}^{(i)} = \frac{p_{\text{tar}}(\mathbf{x}_{t-\Delta t}|\mathbf{x}_t)}{q(\mathbf{x}_{t-\Delta t}|\mathbf{x}_t)} w_t^{(i)} \tag{28}$$

where $p_{\text{tar}}$ is an intermediate target kernel we want to sample from, and $q(\mathbf{x}_{t-\Delta t}|\mathbf{x}_t)$ is a proposal kernel used during propagation.

At each time $t$, if the effective sample size (ESS), defined as $\left(\sum_{j=1}^K w_t^{(j)}\right)^2 / \sum_{i=1}^K \left(w_t^{(i)}\right)^2$ falls below a predefined threshold, resampling is performed. Specifically, a set of ancestor indices $\{a_t^{(i)}\}_{i=1}^K$ is drawn from a multinomial distribution based on the normalized weights. These indices are then used to form the resampled particle set $\{\mathbf{x}_t^{(a_t^{(i)})}\}_{i=1}^K$. If resampling is not triggered, we simply set $a_t^{(i)} = i$.

In the propagation stage, particle set $\{\mathbf{x}_{t-\Delta t}^{(i)}\}_{i=1}^K$ is generated via sampling from proposal distribution, $\mathbf{x}_{t-\Delta t}^{(i)} \sim q(\mathbf{x}_{t-\Delta t}|\mathbf{x}_t^{(a_t^{(i)})})$. When resampling is applied, the weights are reset to uniform values, $i.e.$, $w_t^{(i)} = 1$ for all $i$. Regardless of whether resampling occurred, new weights $\{w_{t-\Delta t}^{(i)}\}_{i=1}^K$ are then computed using Eq. 28.

In the context of reward-alignment tasks, SMC can be employed to approximately sample from the target distribution defined in Eq. 3. As the number of particles $K$ grows, the approximation becomes increasingly accurate due to the consistency of the SMC framework [61, 62]. To make this effective, the proposal kernel should ideally match the optimal transition kernel given in Eq. 23. However, as discussed in Appendix A, this kernel is computationally intractable. Therefore, prior work [14, 12, 8] typically resorts to its approximated form, as expressed in Eq. 27. This leads to the weight at each time being computed as:

$$w_{t-\Delta t}^{(i)} = \frac{\tilde{p}_\theta^*(\mathbf{x}_{t-\Delta t}|\mathbf{x}_t)}{q(\mathbf{x}_{t-\Delta t}|\mathbf{x}_t)} w_t^{(i)} = \frac{\exp(r(\mathbf{x}_{0|t-\Delta t})/\alpha) p_\theta(\mathbf{x}_{t-\Delta t}|\mathbf{x}_t)}{\exp(r(\mathbf{x}_{0|t})/\alpha) q(\mathbf{x}_{t-\Delta t}|\mathbf{x}_t)} w_t^{(i)}, \tag{29}$$

where $p_\theta(\mathbf{x}_{t-\Delta t}|\mathbf{x}_t)$ denotes the transition kernel of the pretrained score-based generative model. The pretrained model follows the SDE given in Eq. 1, which upon discretization yields a Gaussian transition kernel, $p_\theta(\mathbf{x}_{t-\Delta t}|\mathbf{x}_t) = \mathcal{N}(\mathbf{x}_t - \mathbf{f}(\mathbf{x}_t, t)\Delta t, g(t)^2 \Delta t \mathbf{I})$. On the other hand, for reward-guided sampling, $i.e.$, to sample from the target distribution in Eq. 3, we follow controlled SDE in Eq. 14. At each intermediate time, the SOC framework (Appendix A) prescribes the use of the optimal control defined in Eq. 20. However, due to its intractability, the approximation in Eq. 26 is typically adopted in practice. Discretizing the controlled SDE under this approximation leads to the following proposal distribution at each time:

$$q(\mathbf{x}_{t-\Delta t}|\mathbf{x}_t) = \mathcal{N}(\mathbf{x}_t - \mathbf{f}(\mathbf{x}_t, t)\Delta t + g^2(t)\nabla \frac{r(\mathbf{x}_{0|t})}{\alpha}\Delta t, \ g(t)^2 \Delta t \mathbf{I}). \tag{30}$$

A similar proposal has also been used in [12, 8], where a Taylor expansion was applied in the context of entropy-regularized Markov Decision Process.

## C  Acceptance Probability of MALA and pCNL

In this section we provide the Metropolis–Hastings (MH) [69, 30] acceptance rule that underpins both the Metropolis-Adjusted Langevin Algorithm (MALA) [31, 32] and the preconditioned Crank–Nicolson Langevin algorithm (pCNL) [36, 37]. Metropolis-Hastings algorithms form a class of MCMC methods that generate samples from a target distribution by accepting or rejecting proposed moves according to a specific acceptance function. Let $p_{\text{tar}}$ denote a density proportional to the target distribution. Given the current state $\mathbf{x}$ and a proposal $\mathbf{x}' \sim q(\mathbf{x}'|\mathbf{x})$, the MH step accepts the move with probability:

$$a(\mathbf{x}, \mathbf{x}') = \min\left(1, \frac{p_{\text{tar}}(\mathbf{x}')q(\mathbf{x}|\mathbf{x}')}{p_{\text{tar}}(\mathbf{x})q(\mathbf{x}'|\mathbf{x})}\right). \tag{31}$$

If the proposal kernel $q(\mathbf{x}'|\mathbf{x})$ is taken to be the one–step Euler–Maruyama discretization of Langevin dynamics then it becomes the MALA, and Eq. 31 corresponds to the acceptance probability of

MALA. Choosing instead the semi-implicit (Crank–Nicolson-type) discretization yields the proposal used in the pCNL, and Eq. 31 becomes the corresponding pCNL acceptance probability.

We first show that preconditioned Crank-Nicolson (pCN), which is a modification of the Random-Walk Metropolis (RWM), preserves the Gaussian prior. pCN can be viewed as a special case of pCNL obtained when the underlying Langevin dynamics is chosen so that the Gaussian prior $\mathcal{N}(\mathbf{0}, \mathbf{I})$ is its invariant distribution. This leads to the proposal mechanism [36, 37]:

$$\mathbf{x}' = \rho\mathbf{x} + \sqrt{1 - \rho^2}\mathbf{z}, \qquad \mathbf{z} \sim \mathcal{N}(\mathbf{0}, \mathbf{I}). \tag{32}$$

where $\rho = (1 - \epsilon/4)/(1 + \epsilon/4)$ with $\epsilon > 0$ corresponding to the step size of the Langevin dynamics. Assume that the prior is the standard Gaussian $\mathcal{N}(\mathbf{0}, \mathbf{I})$ and let $\mathbf{x} \sim \mathcal{N}(\mathbf{0}, \mathbf{I})$, then Eq. 32 expresses $\mathbf{x}'$ as a linear combination of two independent Gaussian random variables with unit covariance. Hence, by the closure of the Gaussian family under affine transformations, $\mathbf{x}' \sim \mathcal{N}(\mathbf{0}, \mathbf{I})$ as well, thus preserving the Gaussian prior.
Next, in the case of pCN, $p_1(\mathbf{x}')q_0(\mathbf{x}|\mathbf{x}')$, is symmetric, $i.e.$, $p_1(\mathbf{x}')q_0(\mathbf{x}|\mathbf{x}') = p_1(\mathbf{x})q_0(\mathbf{x}'|\mathbf{x})$, where $p_1(\cdot)$ denotes Gaussian prior and $q_0(\cdot|\cdot)$ denotes the proposal kernel of the pCN, $i.e.$, Eq. 32.

**Remark 1.** *Let $p_1(\cdot)$ as Gaussian prior $\mathcal{N}(\mathbf{0}, \mathbf{I})$ and $q_0(\cdot|\cdot)$ as the proposal kernel of the pCN, i.e., $\mathcal{N}(\rho\mathbf{x}, (1 - \rho^2)\mathbf{I})$, then $p_1(\mathbf{x}')q_0(\mathbf{x}|\mathbf{x}') = p_1(\mathbf{x})q_0(\mathbf{x}'|\mathbf{x})$.*

*Proof.* Apart from normalization constants, $p_1(\mathbf{x}')q_0(\mathbf{x}|\mathbf{x}')$ can be calculated as:

$$\exp\left(-\frac{\mathbf{x}'^2}{2}\right)\exp\left(-\frac{(\mathbf{x} - \rho\mathbf{x}')^2}{2(1 - \rho^2)}\right) = \exp\left(-\frac{\mathbf{x}'^2 + \mathbf{x}^2 - 2\rho\mathbf{x}\mathbf{x}'}{2(1 - \rho^2)}\right).$$

Repeating the same calculation with $p_1(\mathbf{x})q_0(\mathbf{x}'|\mathbf{x})$ merely swaps $\mathbf{x}$ and $\mathbf{x}'$, leaving the numerator unchanged. Hence the two products are identical. $\square$

We provide additional remark for ease of calculation.

**Remark 2.** *Let $\mathcal{N}(\mathbf{x}; \boldsymbol{\mu}, \mathbf{C})$ be the density of a multivariate Gaussian with mean $\boldsymbol{\mu}$ and positive-definite covariance $\mathbf{C}$. For fixed $\mathbf{C}$, the ratio of two such densities that differ only in the mean is*

$$\frac{\mathcal{N}(\mathbf{x}; \boldsymbol{\mu}, \mathbf{C})}{\mathcal{N}(\mathbf{x}; \mathbf{0}, \mathbf{C})} = \exp\left(-\frac{1}{2}\|\boldsymbol{\mu}\|_{\mathbf{C}}^2 + \langle\boldsymbol{\mu}, \mathbf{x}\rangle_{\mathbf{C}}\right)$$

*where $\|\boldsymbol{\mu}\|_{\mathbf{C}}^2 := \boldsymbol{\mu}^\top\mathbf{C}^{-1}\boldsymbol{\mu}$ and $\langle\boldsymbol{\mu}, \mathbf{x}\rangle_{\mathbf{C}} := \boldsymbol{\mu}^\top\mathbf{C}^{-1}\mathbf{x}$.*

In our case, $\mathbf{C}$ corresponds to the identity matrix.

**Acceptance Probability of pCNL.** As before, let $q_0$ be a proposal kernel of the pCN (Eq. 32) and $q_p$ be a proposal kernel of the pCNL:

$$q_0(\mathbf{x}'|\mathbf{x}) : \mathbf{x}' = \rho\mathbf{x} + \sqrt{1 - \rho^2}\mathbf{z}, \qquad \mathbf{z} \sim \mathcal{N}(\mathbf{0}, \mathbf{I}) \tag{33}$$

$$q_p(\mathbf{x}'|\mathbf{x}) : \mathbf{x}' = \rho\mathbf{x} + \sqrt{1 - \rho^2}\left(\mathbf{z} + \frac{\sqrt{\epsilon}}{2}\nabla\frac{r(\mathbf{x}_{0|1})}{\alpha}\right), \qquad \mathbf{z} \sim \mathcal{N}(\mathbf{0}, \mathbf{I}). \tag{34}$$

Let $\tilde{\mathbf{x}} := \frac{\mathbf{x}' - \rho\mathbf{x}}{\sqrt{1 - \rho^2}}$, and $\tilde{q}_0, \tilde{q}_p$ be the distributions of $\tilde{\mathbf{x}}$ under $q_0$ and $q_p$, respectively. Then we obtain:

$$\tilde{q}_0(\tilde{\mathbf{x}}|\mathbf{x}) = \mathcal{N}(\tilde{\mathbf{x}}; \mathbf{0}, \mathbf{I}) \tag{35}$$

$$\tilde{q}_p(\tilde{\mathbf{x}}|\mathbf{x}) = \mathcal{N}\left(\tilde{\mathbf{x}}; \frac{\sqrt{\epsilon}}{2}\nabla\frac{r(\mathbf{x}_{0|1})}{\alpha}, \mathbf{I}\right). \tag{36}$$

Note that $\mathbf{x}_{0|1}$ is a function of $\mathbf{x}$. Then by Remark 2,

$$\frac{q_p(\mathbf{x}'|\mathbf{x})}{q_0(\mathbf{x}'|\mathbf{x})} = \frac{\tilde{q}_p(\tilde{\mathbf{x}}|\mathbf{x})}{\tilde{q}_0(\tilde{\mathbf{x}}|\mathbf{x})} = \exp\left(-\frac{\epsilon}{8}\left\|\nabla\frac{r(\mathbf{x}_{0|1})}{\alpha}\right\|_{\mathbf{I}}^2 + \frac{\sqrt{\epsilon}}{2}\left\langle\nabla\frac{r(\mathbf{x}_{0|1})}{\alpha}, \tilde{\mathbf{x}}\right\rangle_{\mathbf{I}}\right). \tag{37}$$

For the fraction part of the acceptance probability (Eq. 31) of pCNL, we have:

$$\frac{p_1^*(\mathbf{x}')q_p(\mathbf{x}|\mathbf{x}')}{p_1^*(\mathbf{x})q_p(\mathbf{x}'|\mathbf{x})} = \frac{(p_1^*(\mathbf{x}')q_p(\mathbf{x}|\mathbf{x}'))/(p_1(\mathbf{x}')q_0(\mathbf{x}|\mathbf{x}'))}{(p_1^*(\mathbf{x})q_p(\mathbf{x}'|\mathbf{x}))/(p_1(\mathbf{x}')q_0(\mathbf{x}|\mathbf{x}'))} \tag{38}$$

$$= \frac{(p_1^*(\mathbf{x}')q_p(\mathbf{x}|\mathbf{x}'))/(p_1(\mathbf{x}')q_0(\mathbf{x}|\mathbf{x}'))}{(p_1^*(\mathbf{x})q_p(\mathbf{x}'|\mathbf{x}))/(p_1(\mathbf{x})q_0(\mathbf{x}'|\mathbf{x}))} := \frac{\varphi_p(\mathbf{x}',\mathbf{x})}{\varphi_p(\mathbf{x},\mathbf{x}')}, \tag{39}$$

where the target distribution is set as Eq. 25. In Eq. 38, we divide both numerator and denominator by a common term, and in Eq. 39, we utilized Remark 1.
Denominator can be calculated utilizing Eq. 37:

$$\varphi_p(\mathbf{x},\mathbf{x}') = \frac{p_1^*(\mathbf{x})q_p(\mathbf{x}'|\mathbf{x})}{p_1(\mathbf{x})q_0(\mathbf{x}'|\mathbf{x})} \tag{40}$$

$$= \exp\left(\frac{r(\mathbf{x}_{0|1})}{\alpha}\right)\exp\left(-\frac{\epsilon}{8}\left\|\nabla\frac{r(\mathbf{x}_{0|1})}{\alpha}\right\|_{\mathbf{I}}^2 + \frac{\sqrt{\epsilon}}{2}\left\langle\nabla\frac{r(\mathbf{x}_{0|1})}{\alpha}, \frac{\mathbf{x}'-\rho\mathbf{x}}{\sqrt{1-\rho^2}}\right\rangle_{\mathbf{I}}\right), \tag{41}$$

with numerator being simply interchanging $\mathbf{x}$ and $\mathbf{x}'$. The acceptance probability of pCNL is $\min\left(1, \frac{\varphi_p(\mathbf{x}',\mathbf{x})}{\varphi_p(\mathbf{x},\mathbf{x}')}\right)$.

**Acceptance Probability of MALA.**  In the case of MALA, the proposal is given as:

$$\mathbf{x}' = \mathbf{x} + \frac{\epsilon}{2}\nabla\log p_{\text{tar}}(\mathbf{x}) + \sqrt{\epsilon}\mathbf{z}, \qquad \mathbf{z} \sim \mathcal{N}(\mathbf{0},\mathbf{I}) \tag{42}$$

$$= \mathbf{x} + \frac{\epsilon}{2}\left(-\mathbf{x} + \nabla\frac{r(\mathbf{x}_{0|1})}{\alpha}\right) + \sqrt{\epsilon}\mathbf{z}, \qquad \mathbf{z} \sim \mathcal{N}(\mathbf{0},\mathbf{I}) \tag{43}$$

where as in pCNL we set the target distribution as Eq. 25. Thus the proposal kernel of MALA $q_M$ can be expressed as:

$$q_M(\mathbf{x}'|\mathbf{x}) = \mathcal{N}\left(\mathbf{x}'; \mathbf{x}\left(1-\frac{\epsilon}{2}\right) + \frac{\epsilon}{2}\nabla\frac{r(\mathbf{x}_{0|1})}{\alpha}, \epsilon\mathbf{I}\right). \tag{44}$$

The fraction part of the acceptance probability (Eq. 31) of MALA is given as:

$$\frac{p_1^*(\mathbf{x}')q_M(\mathbf{x}|\mathbf{x}')}{p_1^*(\mathbf{x})q_M(\mathbf{x}'|\mathbf{x})} = \frac{\mathcal{N}(\mathbf{x}'; \mathbf{0},\mathbf{I})\exp(r(\mathbf{x}'_{0|1})/\alpha)\mathcal{N}(\mathbf{x}; \mathbf{x}'(1-\epsilon/2)+\epsilon/2\cdot\nabla r(\mathbf{x}'_{0|1})/\alpha,\epsilon\mathbf{I})}{\mathcal{N}(\mathbf{x}; \mathbf{0},\mathbf{I})\exp(r(\mathbf{x}_{0|1})/\alpha)\mathcal{N}(\mathbf{x}'; \mathbf{x}(1-\epsilon/2)+\epsilon/2\cdot\nabla r(\mathbf{x}_{0|1})/\alpha,\epsilon\mathbf{I})} \tag{45}$$

$$:= \frac{\varphi_M(\mathbf{x}',\mathbf{x})}{\varphi_M(\mathbf{x},\mathbf{x}')} \tag{46}$$

where we denote $\mathbf{x}'_{0|1} := \mathbf{x}_{0|1}(\mathbf{x}')$, *i.e.*, we calculate Tweedie's formula with $\mathbf{x}'$. Thus the denominator, which is $\varphi_M(\mathbf{x},\mathbf{x}')$, is proportional to the following expression:

$$\exp\left(-\frac{\|\mathbf{x}\|_{\mathbf{I}}^2}{2} + \frac{r(\mathbf{x}_{0|1})}{\alpha} - \frac{\|\mathbf{x}'-\{\mathbf{x}(1-\frac{\epsilon}{2})+\frac{\epsilon}{2}\nabla\frac{r(\mathbf{x}_{0|1})}{\alpha}\}\|_{\mathbf{I}}^2}{2\epsilon}\right). \tag{47}$$

After simplifying the expression—specifically, canceling out the cross terms involving $\mathbf{x}$ and $\mathbf{x}'$ that will appear symmetrically in the numerator and denominator—the expression for $\varphi_M(\mathbf{x},\mathbf{x}')$ becomes:

$$\varphi_M(\mathbf{x},\mathbf{x}') \tag{48}$$

$$= \exp\left(\frac{r(\mathbf{x}_{0|1})}{\alpha}\right)\exp\left(-\frac{\epsilon}{8}\left\|\nabla\frac{r(\mathbf{x}_{0|1})}{\alpha}\right\|_{\mathbf{I}}^2 - \frac{\epsilon}{8}\|\mathbf{x}\|_{\mathbf{I}}^2 + \frac{1}{2}\left\langle\nabla\frac{r(\mathbf{x}_{0|1})}{\alpha}, \left(\mathbf{x}'-\left(1-\frac{\epsilon}{2}\right)\mathbf{x}\right)\right\rangle_{\mathbf{I}}\right). \tag{49}$$

The numerator $\varphi_M(\mathbf{x}',\mathbf{x})$ can be obtained by simply interchanging $\mathbf{x}$ and $\mathbf{x}'$. The acceptance probability of MALA is then, $\min\left(1, \frac{\varphi_M(\mathbf{x}',\mathbf{x})}{\varphi_M(\mathbf{x},\mathbf{x}')}\right)$.

# D   Experimental Setup and Details

In this section, we provide comprehensive details for each application: layout-to-image generation, quantity-aware image generation, and aesthetic-preference image generation. We also include full experimental details.

**Layout-to-Image Generation.**   This task involves placing user-specified objects within designated bounding boxes [82–85]. We evaluate performance on 50 randomly sampled cases from the HRS-Spatial [80] dataset. As the reward model, we use GroundingDINO [72], and measure the alignment between predicted and target boxes using mean Intersection-over-Union (mIoU). For the held-out reward, we compute mIoU using a different object detector, Salience DETR [74].

**Quantity-Aware Image Generation.**   This task involves generating a user-specified object in a specified quantity [86–88]. We evaluate methods on a custom dataset constructed via GPT-4o [81], comprising 20 object categories with randomly assigned counts up to 90, totaling 40 evaluation cases. As the reward model, we used T2ICount [73], which takes a generated image and the corresponding text prompt as input and returns a density map. Summing over this density map yields a differentiable estimate of the object count $n_{\text{pred}}$. The reward is defined as the negative smooth L1 loss:

$$r_{\text{count}} = \begin{cases} -0.5(n_{\text{pred}} - n_{\text{gt}})^2 & |n_{\text{pred}} - n_{\text{gt}}| < 1, \\ -|n_{\text{pred}} - n_{\text{gt}}| + 0.5 & |n_{\text{pred}} - n_{\text{gt}}| \geq 1. \end{cases}$$

where $n_{\text{gt}}$ denotes the input object quantity. For the held-out reward, we used an alternative counting model, CountGD [75]. This model returns integer-valued object counts. We apply a confidence threshold of 0.3 and evaluate using mean absolute error (MAE) and counting accuracy, where a prediction is considered correct if $n_{\text{pred}} = n_{\text{gt}}$.

**Aesthetic-Preference Image Generation**   This task involves generating visually appealing images. We evaluate performance using 45 prompts consisting of animal names, provided in [45]. As the reward model, we use the LAION Aesthetic Predictor V2 [76], which estimates the aesthetic quality of an image and is commonly used in reward-alignment literature [12, 51, 60, 45].

**Common Held-Out Reward Models.**   For all applications, we additionally evaluate the generated images using widely adopted held-out reward models for image quality and text alignment. Specifically, we use ImageReward [77], fine-tuned on human feedback, and VQAScore [78], which leverages a visual question answering (VQA) model. Both are based on vision-language models and assess how well the generated image aligns with the input text prompt.

**Experimental Details.**   We use FLUX-Schnell [28] as the score-based generative model for our method and all baselines. Although FLUX is a flow-based model, SMC-based inference-time reward alignment can be applied by reformulating the generative ODE as an SDE [7, 13], as described in Sec. 3.1. Further to ensure diversity between samples during SMC, we applied Variance Preserving (VP) interpolant conversion [13, 54]. Apart from FLUX, we additionally report quantitative and qualitative results using another score-based generative model, SANA-Sprint [29], in Appendix H. These results demonstrate that the our claims are not tied to a specific model architecture, and additionally highlighting the robustness of $\Psi$-SAMPLER.

We use 25 denoising steps for all SMC-based methods and 50 for single-particle methods to compensate for their reduced exploration capacity. To ensure fair comparison, we fix the total number of function evaluations (NFE) across all SMC variants—1,000 for layout-to-image and quantity-aware generation tasks, and 500 for aesthetic-preference image generation. For aesthetic-preference image generation, we used half the NFE compared to other tasks because we found it sufficient for performance convergence. For methods that sample from the posterior, half of the NFE is allocated to the initial sampling stage, resulting in 20 particles during SMC, while prior-based methods use all NFE for SMC with 40 particles. For the aesthetic-preference image generation, we use half the number of particles in both settings to reflect the halved NFE. The ablation study comparing the performance on varying NFE allocation is provided in Appendix G. For all experiments, we used NVIDIA A6000 GPUs with 48GB VRAM.

# E   Toy Experiment Setup and Details

In this section, we describe the setup for the toy experiment presented in the main paper. Specifically, the data distribution $p_0$ is a six-mode Gaussian mixture with covariance $0.3\mathbf{I}$, consisting of

six equally weighted components uniformly arranged on a circle of radius 6. The reward function is defined as the sum of three Gaussian components centered at points evenly spaced on the circle: $r(x) = \sum_{i=1}^{3} \exp\left(-0.5\|x - p_i\|^2\right)$, where $p_1 = (6.5, 0)$, $p_1 = (-3.25, 3.25\sqrt{3})$, and $p_3 = (-3.25, -3.25\sqrt{3})$. We trained a few-step score-based generative model using 4-layer MLP with hidden dimension 128. For sampling, we fixed the total NFE to 100 across all methods and visualized the results using 2,000 generated samples. We used step size 0.1 for MALA and 0.2 for pCNL. We deliberately used a smaller step size for MALA than for pCNL to better reflect the actual experimental settings used in the main experiments (Sec. 5.1).

## F Additional Evaluation Results of MALA and pCNL Initializations under Varying Step Size

We presented a comparison between MALA and pCNL under varying MCMC step sizes for the layout-to-image generation task in the Sec. 5.4. Here, we extend this analysis to the remaining two tasks. In Fig. 4, we report results for quantity-aware generation (top row) and aesthetic-preference generation (bottom row), comparing MALA and pCNL across a range of step sizes.
For the quantity-aware image generation task, we report acceptance probability, reward (negative smooth L1 loss via T2I-Count[73]), held-out reward (mean absolute error (MAE) via CountGD [75]), and LPIPS Mean Pairwise Distance (MPD) [79] (which measures the sample diversity). For the aesthetic-preference task, we report acceptance probability, reward (aesthetic score [76]), and LPIPS MPD [79]. All metrics are directly computed from the Tweedie estimates [18] of MCMC samples, before the SMC stage.

As the step size increases, MALA's acceptance probability quickly falls to near zero, whereas pCNL retains a stable acceptance rate across a wider range. In quantity-aware image generation, pCNL achieves its best T2I-Count reward and lowest MAE at moderate step sizes (approximately 0.5–1.0), while MALA's performance deteriorates sharply beyond 0.05. Although pCNL's LPIPS MPD decreases at larger step sizes, it consistently outperforms MALA in diversity across the same settings. In aesthetic-preference generation, pCNL generally achieves higher aesthetic scores, with only a slight drop at the smallest step size, and maintains higher LPIPS MPD across all settings. In contrast, MALA's aesthetic scores decline rapidly once its acceptance rate vanishes.

These results demonstrate that pCNL can effectively leverage larger step sizes to improve sample quality, with only minimal trade-offs in diversity.

### Quantity-aware Generation

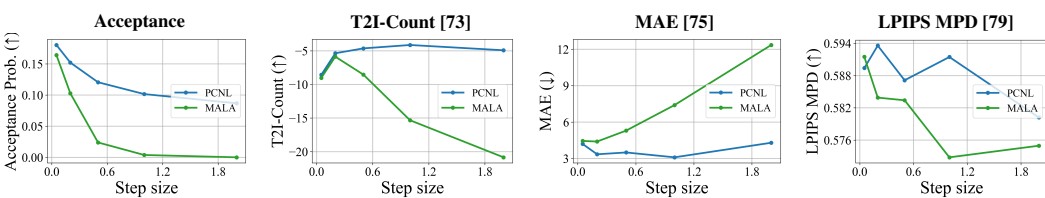

### Aesthetic-preference Generation

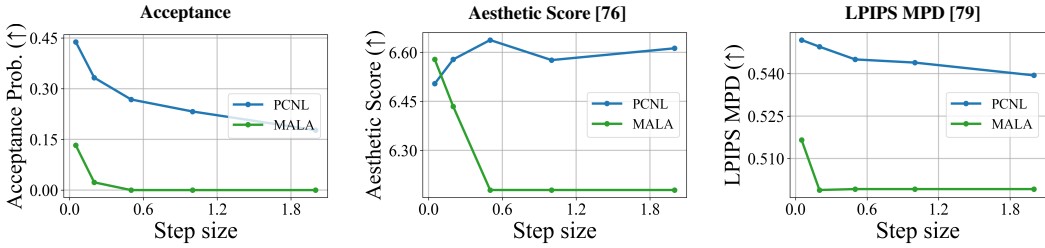

Figure 4: Performance comparison of MALA and pCNL across different evaluation metrics under two generation settings: *quantity-aware generation* (top row) and *aesthetic-preference generation* (bottom row). Each graph illustrates the performance trend with varying step sizes.

# G   Ablation Study: Varying NFE Allocation for Initial Particle Sampling Stage

In this section, we present an ablation study examining how performance varies with different allocations of the total number of function evaluations (NFE) between the initial particle sampling stage (via Top-$K$-of-$N$ or MCMC) and the subsequent SMC stage. In our main experiments, we adopt a balanced allocation, with 50% of the NFE budget used for initial particle sampling and the remaining 50% for SMC (denoted as 50%/50%).

To assess the effect of this design choice, we evaluate two alternative NFE splits: 25% for initial particle sampling and 75% for SMC (denoted as 25%/75%), and 75% for initial particle sampling with only 25% for SMC (denoted as 75%/25%). We conduct this analysis on both Top-$K$-of-$N$ and $\Psi$-SAMPLER. For consistency, we fix the number of SMC steps to 25, adjusting the number of particles accordingly: 30 particles for the 25%/75% setting and 10 particles for the 75%/25% setting. For the aesthetic-preference image generation, we use half the number of particles in both settings to reflect the halved NFE. Tab. 2 summarizes the results.

The 75%/25% split overinvests compute to initial particle sampling, resulting in inefficiency due to the significantly reduced number of particles used during the SMC phase. As a result, it consistently showed the worst performance across all reward metrics. Conversely, the 25%/75% split dedicates too little budget to initial particle sampling, limiting exploration despite using 1.5 times more particles than the 50%/50% split. This leads to weaker performance, particularly under held-out reward evaluations. In contrast, the balanced 50%/50% split consistently yields the most robust performance across both seen and held-out rewards across all tasks.

| Tasks | Metrics | Top-$K$-of-$N$ | | | $\Psi$-SAMPLER | | |
|---|---|---|---|---|---|---|---|
| | | 25%/75% | 50%/50% | 75%/25% | 25%/75% | 50%/50% | 75%/25% |
| **Layout to Image** | GroundingDINO[†] [72] ↑ | 0.424 | 0.425 | 0.390 | 0.454 | **0.467** | 0.433 |
| | mIoU [74] ↑ | 0.439 | 0.427 | 0.401 | 0.463 | **0.471** | 0.426 |
| | ImageReward [77] ↑ | **1.142** | 0.957 | 0.913 | 1.128 | 1.035 | 0.884 |
| | VQA [78] ↑ | 0.822 | **0.855** | 0.770 | 0.825 | 0.810 | 0.766 |
| **Quantity Aware** | T2I-Count[†] [73] ↓ | 1.021 | 1.077 | 2.934 | **0.804** | 0.850 | 1.892 |
| | MAE [75] ↓ | 4.6 | 3.675 | 5.65 | 3.6 | **2.925** | 3.7 |
| | Acc (%) [75] ↑ | 22.5 | 12.5 | 25.0 | 25.0 | **32.5** | 30.0 |
| | ImageReward [77] ↑ | 0.739 | 0.752 | 0.714 | 0.693 | **0.796** | 0.694 |
| | VQA [78] ↑ | 0.910 | **0.960** | 0.937 | 0.932 | 0.951 | 0.943 |
| **Aesthetic Preference** | Aesthetic[†] [76] ↑ | 6.958 | 6.879 | 6.853 | **7.015** | 7.012 | 6.868 |
| | ImageReward [77] ↑ | 1.114 | 1.133 | 1.076 | 1.040 | **1.171** | 1.062 |
| | VQA [78] ↑ | 0.964 | 0.961 | **0.969** | 0.968 | 0.963 | 0.952 |

Table 2: Ablation study results on varying NFE allocation conducted with $\Psi$-SAMPLER and Top-$K$-of-$N$ across three task domains. **Bold** indicates the best performance, while underline denotes the second-best result for each metric. Metrics marked with [†] are used as seen reward during reward-guided sampling, where others are held-out reward. Higher values indicate better performance (↑), unless otherwise noted (↓).

# H  Ψ-SAMPLER with Other Score-Based Generative Model

To evaluate the generality of Ψ-SAMPLER and further support our claims, we additionally conduct experiments using SANA-Sprint [29], which is a few-step flow-based generative models.

Despite architectural differences, SANA-Sprint integrates seamlessly with our Ψ-SAMPLER. As shown in Tab. 3, it demonstrates reward and quality improvements consistent with those observed using FLUX [28] in Tab. 1, indicating that our method generalizes beyond a specific backbone. Notably, Ψ-SAMPLER delivers the highest performance across all seen reward models and maintains strong generalization to held-out metrics. Moreover, among SMC-based methods, those that initialize particles from the posterior consistently outperform prior-based variants (except for few case), highlighting the benefit of posterior-informed initialization.

These results further highlight the robustness of Ψ-SAMPLER and its applicability to a various few-step score-based generative models.

We further provide qualitative results for each applications, conducted with SANA-Sprint, in Fig. 5. For the layout-to-image generation task, each example shows the input layout with color-coded phrases and corresponding bounding boxes for clarity. In the quantity-aware image generation task, we overlay the predicted object centroids from a held-out counting model [75] on each image to facilitate comparison. The predicted count along with its absolute difference from the target quantity are shown beneath each image in the format $(\Delta\cdot)$, with the best-performing result highlighted in blue. Note that for the aesthetic-preference generation task, the first row corresponds to the prompt "Dog" and the second to "Turkey". We display the generated images alongside their predicted aesthetic scores [76].

| Tasks | Metrics | Single Particle | | SMC-Based Methods | | | | | |
| | | | | Sampling from Prior | | Sampling from Posterior | | | |
| | | DPS [39] | FreeDoM [40] | TDS [14] | DAS [12] | Top-$K$-of-$N$ | ULA | MALA | Ψ-SAMPLER |
|---|---|---|---|---|---|---|---|---|---|
| Layout to Image | GroundingDINO[†] [72] ↑ | 0.144 | 0.159 | 0.403 | 0.338 | 0.406 | 0.388 | 0.392 | **0.429** |
| | mIoU [74] ↑ | 0.229 | 0.242 | 0.405 | 0.343 | 0.406 | 0.393 | 0.394 | **0.432** |
| | ImageReward [77] ↑ | 1.241 | 1.068 | 1.363 | 1.263 | 1.478 | 1.227 | 1.326 | **1.502** |
| | VQA [78] ↑ | 0.779 | 0.754 | 0.835 | 0.808 | 0.851 | 0.776 | 0.832 | **0.853** |
| Quantity Aware | T2I-Count[†] [73] ↓ | 11.290 | 12.839 | 0.110 | 0.122 | 0.0628 | 0.220 | 0.148 | **0.027** |
| | MAE [75] ↓ | 12.3 | 13.8 | 3.475 | 4.55 | 2.825 | 3.025 | 2.375 | **2.175** |
| | Acc (%) [75] ↑ | 0.0 | 0.0 | 30.0 | 22.5 | 27.5 | 22.5 | 30.0 | **32.5** |
| | ImageReward [77] ↑ | 0.680 | 0.526 | **0.954** | 0.889 | 0.803 | 0.789 | 0.840 | 0.845 |
| | VQA [78] ↑ | 0.920 | 0.859 | **0.934** | 0.920 | 0.916 | 0.928 | 0.922 | 0.930 |
| Aesthetic Preference | Aesthetic[†] [76] ↑ | 6.432 | 6.281 | 7.436 | 7.324 | 7.452 | 7.343 | 7.412 | **7.469** |
| | ImageReward [77] ↑ | 1.106 | 1.045 | 1.233 | 1.258 | 1.144 | **1.367** | 1.217 | 1.262 |
| | VQA [78] ↑ | 0.891 | 0.902 | 0.894 | 0.907 | 0.888 | 0.905 | 0.904 | **0.909** |

Table 3: Quantitative comparison of Ψ-SAMPLER and baselines across three task domains, conducted on SANA-Sprint [29]. **Bold** indicates the best performance, while underline denotes the second-best result for each metric. Metrics marked with [†] are used as seen reward during reward-guided sampling, where others are held-out reward. Higher values indicate better performance (↑), unless otherwise noted (↓).

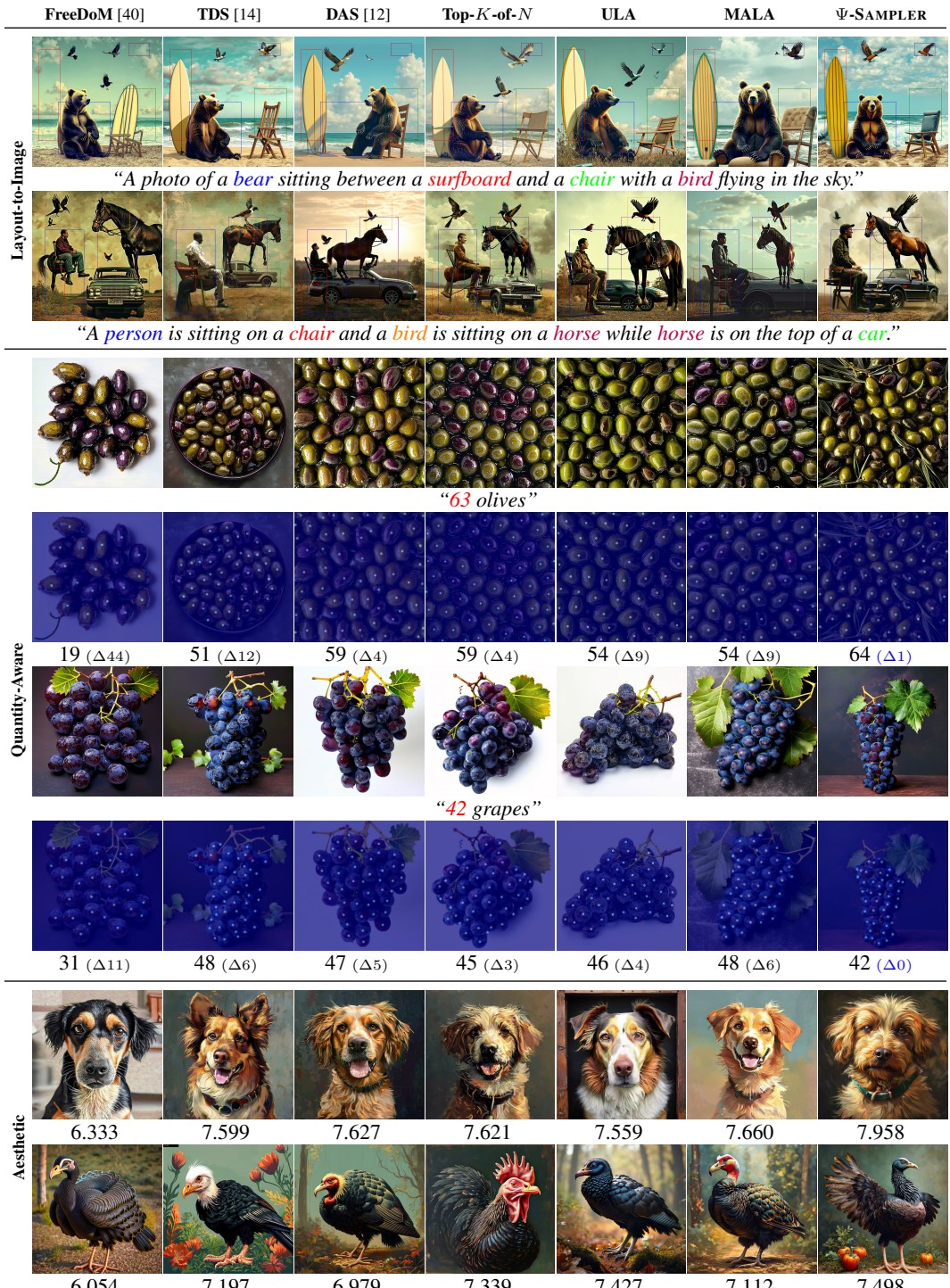

Figure 5: Qualitative results for each application on SANA-Sprint [29].

# I  Additional Qualitative Results

In this section, we present additional qualitative results that extend the examples shown in Fig. 2 of the main paper. Consistent with the Fig. 2, all results here are generated using FLUX [28].

**Layout-to-Image Generation.**    We present additional qualitative results for the layout-to-image generation task in Fig. 6. Each example visualizes the input layout with color-coded phrases and their corresponding bounding boxes for clarity. $\Psi$-SAMPLER consistently respects both the spatial constraints and object presence specified in the layout. The first four rows (Row 1–4) illustrate failure cases where baseline methods generate objects in incorrect locations—either misaligned with the bounding boxes or placed in unrelated regions. For instance, in Row 1, baseline methods fail to accurately place objects within the designated bounding boxes—both the dog and the eagle appear misaligned. Similarly, in Row 4, objects such as the cat and skateboard do not conform to the specified spatial constraints, spilling outside their intended regions in the baseline outputs. In contrast, $\Psi$-SAMPLER successfully generates all objects within their designated bounding boxes. The last four rows (Rows 5–8) illustrate more severe failure cases by baselines, where not only is spatial alignment severely violated, but some objects are entirely missing. For example, in Row 5, DAS [12] fails to generate the apple altogether, while the other baselines exhibit significant spatial misalignment. In Row 7, some baselines produce unrealistic object combinations—such as the apple and red cup being merged—and misinterpret the layout, placing the apple inside the red cup instead of correctly positioning it in the wooden bowl. $\Psi$-SAMPLER not only positions each object correctly but also ensures that all described entities are present and visually distinct.

**Quantity-Aware Image Generation.**    We provide additional qualitative results for the quantity-aware image generation task in Fig. 7 and Fig. 8. The examples cover a variety of object categories and target counts, showing that $\Psi$-SAMPLER works reliably across different scenarios. For each image, we overlay the predicted object centroids from a held-out counting model [75] for easier comparison. Additionally, we display the predicted count below each image, along with the absolute difference from the target quantity in the format $(\Delta \cdot)$. We highlight best case with blue color. $\Psi$-SAMPLER consistently generates the correct number of objects, even in more challenging cases like cluttered scenes or small, overlapping items. On the other hand, baseline methods often produce too many or too few objects, and sometimes include misleading objects. This trend holds across all categories, from food to everyday objects.

**Aesthetic-Preference Image Generation.**    Further qualitative results for aesthetic-preference image generation are presented in Fig. 9. For each prompt (e.g., *"Horse"*, *"Bird"*), we show the predicted aesthetic score [76] below each image. While all methods generate visually plausible outputs, $\Psi$-SAMPLER consistently produces images with higher aesthetic appeal, as reflected in both qualitative impressions and the predicted aesthetic scores.

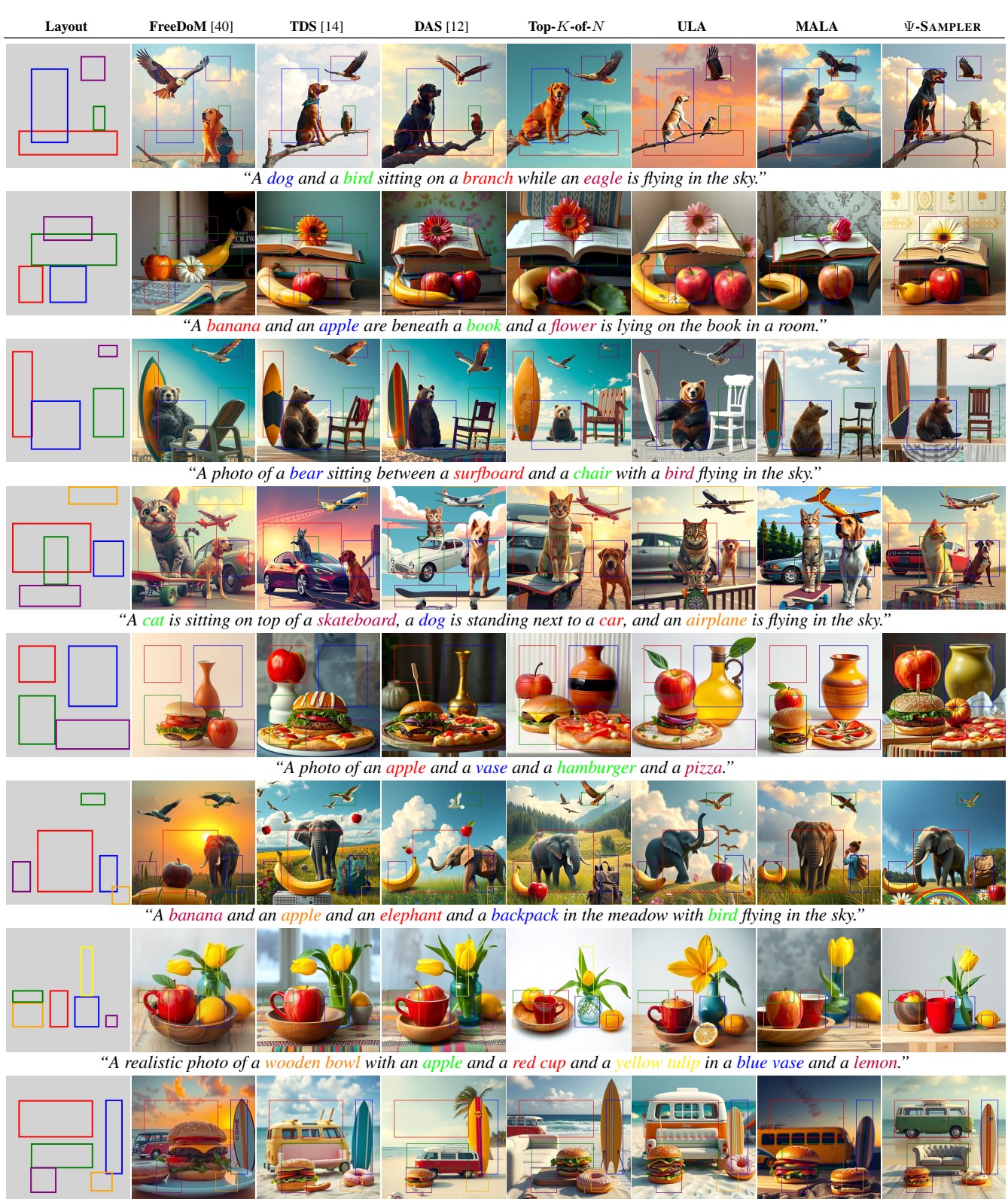

| Layout | FreeDoM [40] | TDS [14] | DAS [12] | Top-$K$-of-$N$ | ULA | MALA | $\Psi$-SAMPLER |
|---|---|---|---|---|---|---|---|

*"A dog and a bird sitting on a branch while an eagle is flying in the sky."*

*"A banana and an apple are beneath a book and a flower is lying on the book in a room."*

*"A photo of a bear sitting between a surfboard and a chair with a bird flying in the sky."*

*"A cat is sitting on top of a skateboard, a dog is standing next to a car, and an airplane is flying in the sky."*

*"A photo of an apple and a vase and a hamburger and a pizza."*

*"A banana and an apple and an elephant and a backpack in the meadow with bird flying in the sky."*

*"A realistic photo of a wooden bowl with an apple and a red cup and a yellow tulip in a blue vase and a lemon."*

*"A realistic photo, a hamburger and a donut and a couch and a bus and a surfboard in the beach."*

Figure 6: **Qualitative results for layout-to-image generation.** Examples show how different methods place objects based on input layouts. $\Psi$-SAMPLER aligns well with the given boxes, while baselines often misplace or miss objects.

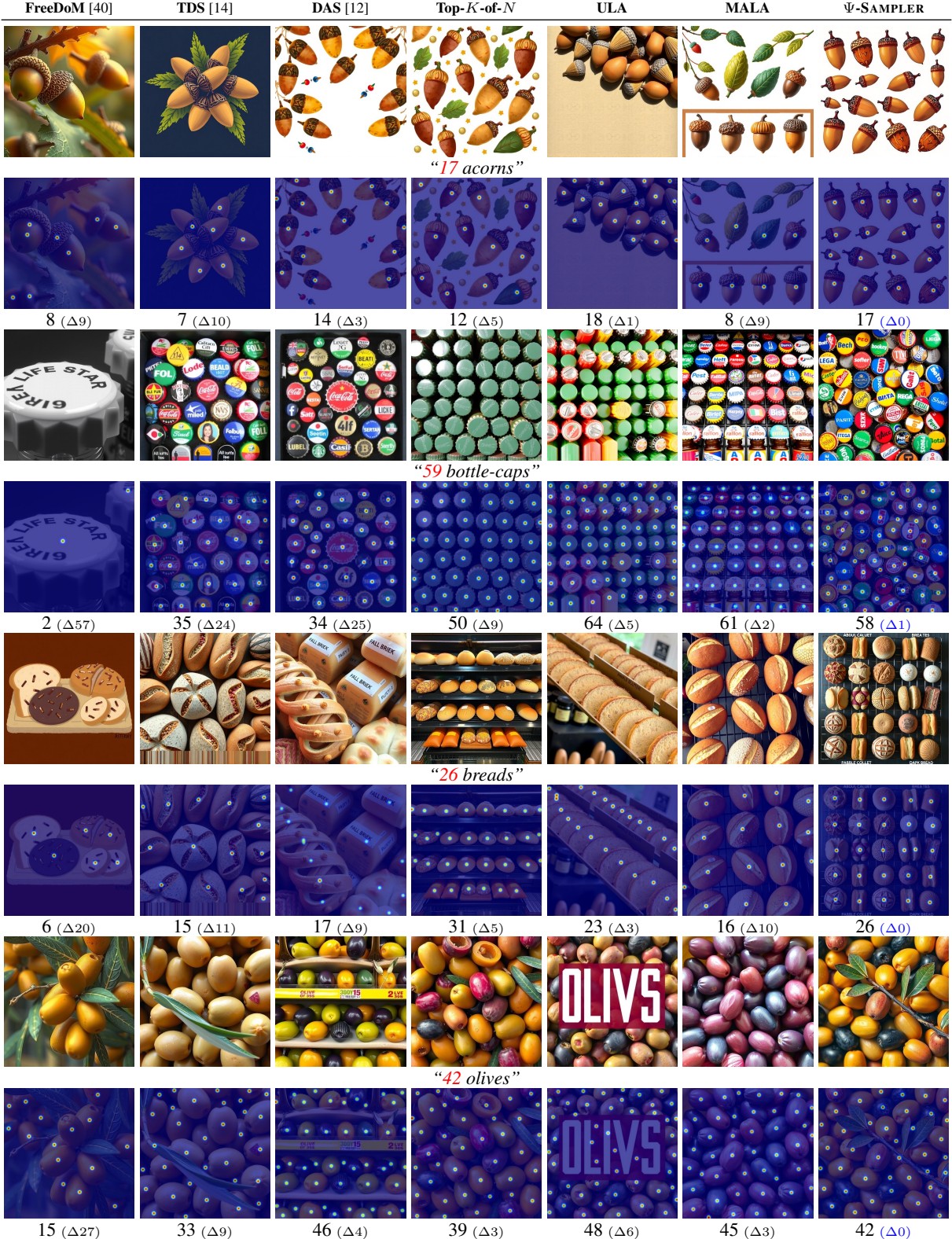

Figure 7: **Qualitative results for quantity-aware image generation.** Across various object types and target counts, $\Psi$-SAMPLER generates the right number of instances more reliably than baseline methods, which tend to over- or under-count.

| FreeDoM [40] | TDS [14] | DAS [12] | Top-$K$-of-$N$ | ULA | MALA | $\Psi$-SAMPLER |
|---|---|---|---|---|---|---|

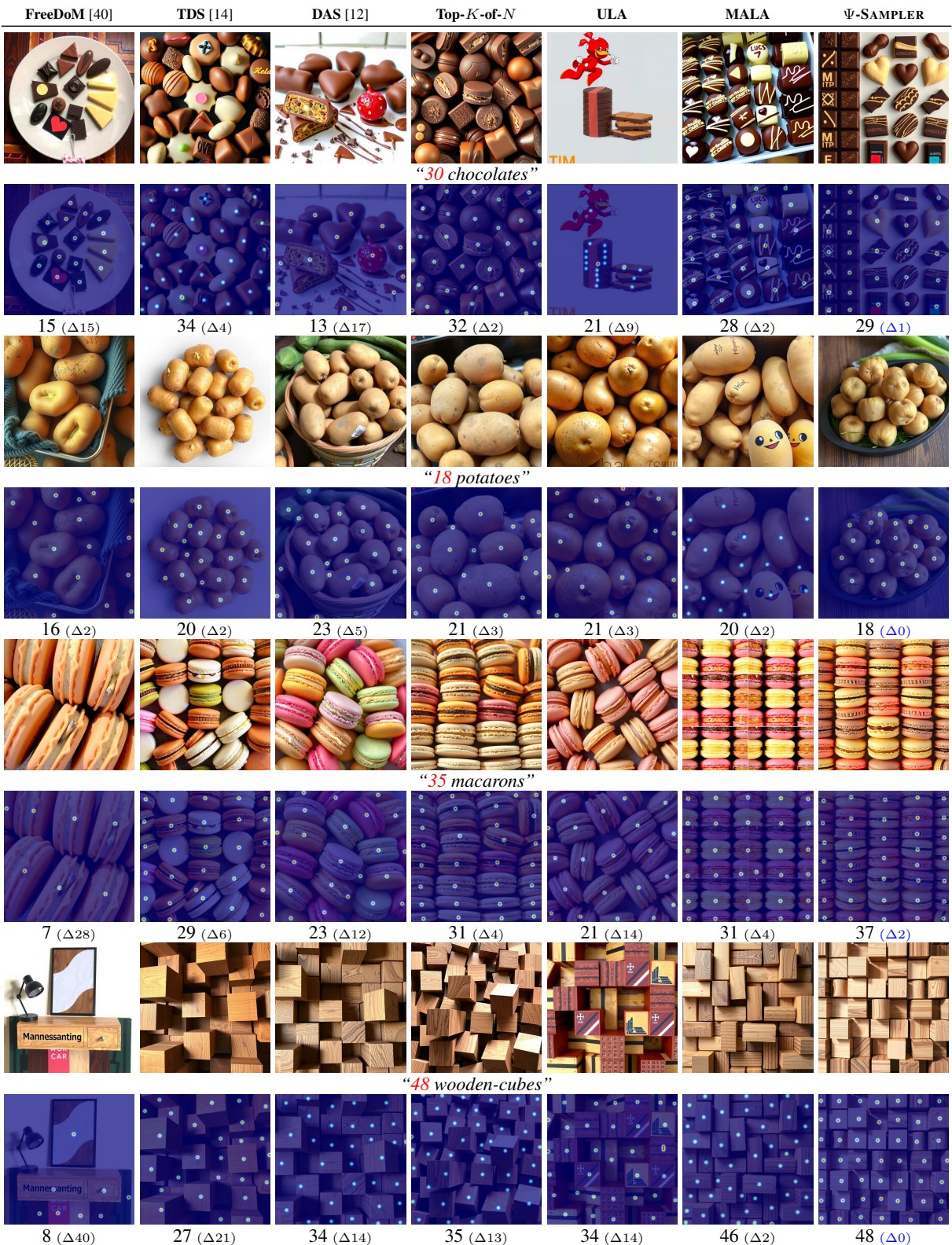

| | | | *"30 chocolates"* | | | |
|---|---|---|---|---|---|---|
| 15 ($\triangle$15) | 34 ($\triangle$4) | 13 ($\triangle$17) | 32 ($\triangle$2) | 21 ($\triangle$9) | 28 ($\triangle$2) | 29 ($\triangle$1) |
| | | | *"18 potatoes"* | | | |
| 16 ($\triangle$2) | 20 ($\triangle$2) | 23 ($\triangle$5) | 21 ($\triangle$3) | 21 ($\triangle$3) | 20 ($\triangle$2) | 18 ($\triangle$0) |
| | | | *"35 macarons"* | | | |
| 7 ($\triangle$28) | 29 ($\triangle$6) | 23 ($\triangle$12) | 31 ($\triangle$4) | 21 ($\triangle$14) | 31 ($\triangle$4) | 37 ($\triangle$2) |
| | | | *"48 wooden-cubes"* | | | |
| 8 ($\triangle$40) | 27 ($\triangle$21) | 34 ($\triangle$14) | 35 ($\triangle$13) | 34 ($\triangle$14) | 46 ($\triangle$2) | 48 ($\triangle$0) |

Figure 8: **Qualitative results for quantity-aware image generation.** Across various object types and target counts, $\Psi$-SAMPLER generates the right number of instances more reliably than baseline methods, which tend to over- or under-count.

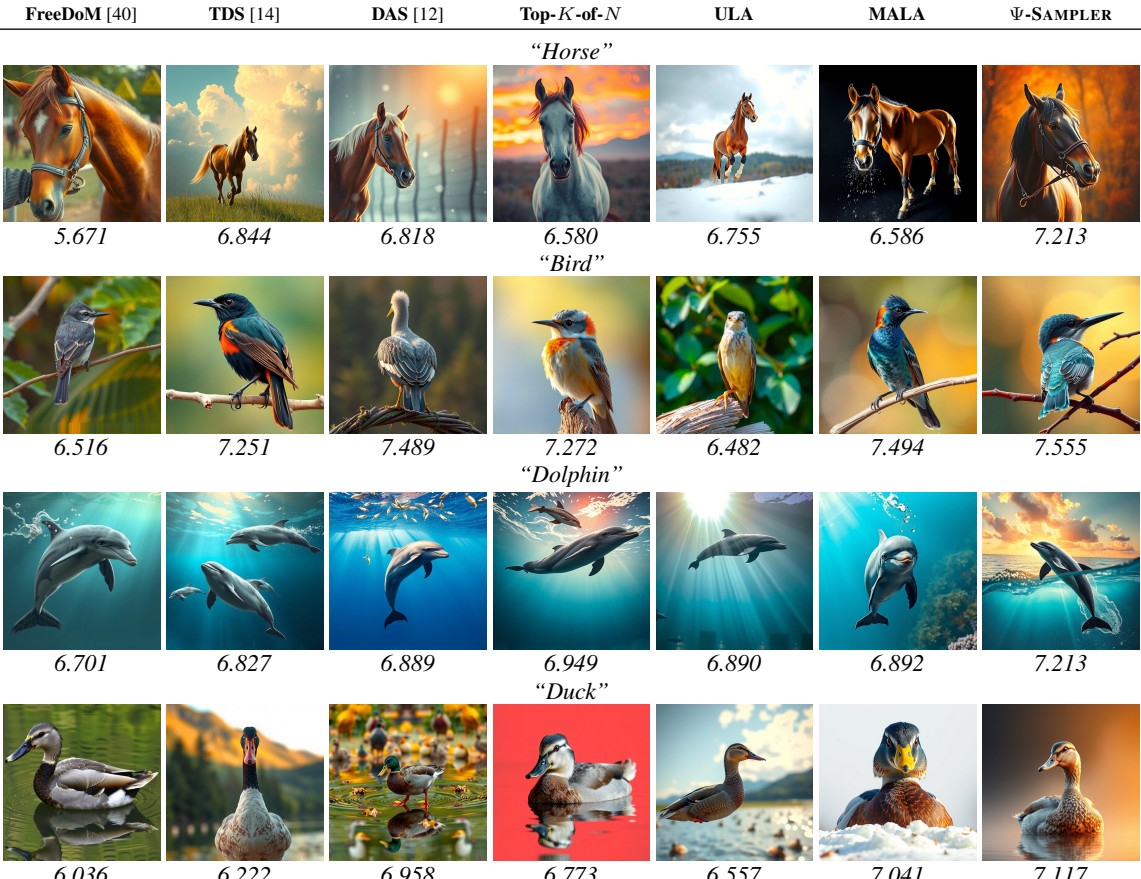

| FreeDoM [40] | TDS [14] | DAS [12] | Top-$K$-of-$N$ | ULA | MALA | $\Psi$-Sampler |
|---|---|---|---|---|---|---|
| | | | *"Horse"* | | | |
| 5.671 | 6.844 | 6.818 | 6.580 | 6.755 | 6.586 | 7.213 |
| | | | *"Bird"* | | | |
| 6.516 | 7.251 | 7.489 | 7.272 | 6.482 | 7.494 | 7.555 |
| | | | *"Dolphin"* | | | |
| 6.701 | 6.827 | 6.889 | 6.949 | 6.890 | 6.892 | 7.213 |
| | | | *"Duck"* | | | |
| 6.036 | 6.222 | 6.958 | 6.773 | 6.557 | 7.041 | 7.117 |

Figure 9: **Qualitative results for aesthetic-preference image generation.** $\Psi$-SAMPLER produces images that are not only realistic but also more visually appealing, with better focus, balance, and overall look compared to baseline outputs.

