# OpenReview forum: "$\Psi$-Sampler: Initial Particle Sampling for SMC-Based Inference-Time Reward Alignment in Score Models"
_NeurIPS.cc/2025/Conference — NeurIPS 2025 spotlight_

### Official Review · Reviewer_bsVN · 2025-06-30

**Clarity:** 2
**Significance:** 3
**Originality:** 3
**Rating:** 5
**Confidence:** 4

**Summary:**

The paper proposes a method to better initialize particle samples for Sequential Monte Carlo samplers used to solve the problem of training-free guidance of diffusion methods. It does so by introducing a MCMC algorithm that targets the tractable version of the posterior distribution at time t=1, obtained through Tweedie's formula. It proposes in particular the pCNL MCMC algorithm to sample from the aforementioned distribution. The numerical experiments show both that there is a real gain of doing so in one toy problem and in several applications using several of the most recent generative models. It also shows that amognst the several candidates samplers that could in theory be used to initialize the distributions, pCNL seems to be the most efficient one.

**Questions:**

1. What happens to MALA in the mixture of Gaussian case?  Seems like an initialization problem, are those points stuck in local Minima? If so, why the pCNL is not stuck in the same local minima? Here the acceptance rate should not have a big importance, as the dimension is low.
2. It should be stated somewhere in the text that the target distribution for the initialization is not exactly the posterior, but it's tractable version (obtained through Tweedie's) somewhere close to equation 11. It is stated that the approximations will be used in equation 3 but still can be misleading by the way equation 11 is written. I suggest the authors either define a notation for the Tweedie approximation that is distinct of the real potential or explicitly say in the text that they will use the tractable counterpart.
3. The part of the toy problem of better mode coverage than MALA does not seem relevant, as both cover all the modes. The problem with MALA seems to be that it adds other modes to the prior due to the initialization.
4. A detailled description of the setting used to produce the toy problem is lacking. It is not currently given in the appendix.

**Ethical Concerns:**

["NO or VERY MINOR ethics concerns only"]

**Final Justification:**

The authors addressed my concerns during the rebuttal phase. As they were mostly related to clarity of the original text and the authors are engaged into adding further explanation and experiments performed during the rebuttal to the paper, I think the paper should be accepted.

**Limitations:**

yes

**Quality:**

3

**Strengths And Weaknesses:**

Strengths:
The paper proposes an original idea to enhance considerably the performance of Sequential Monte Carlo methods for training-free conditional generation in diffusion models. It does so by proposing a suitable MCMC sampler to target the Tweedie's approximation of the posterior distribution at t=1. It evaluates the proposed method in several relevant and nowadays used problems, namely with the latest generative models used.

Weaknesses:
I think that while the evaluation of the current method in industry relevant problems, SMC is a fundamentally probabilistic tool. In this sense, I think that the evaluation in "toy problems" could be further enhanced and clarified. Namely, by running systematic evaluation for several mixtures and several different conditioning problems with a statistical like metric (such as the sliced Wasserstein), one would be more confident in the performance of the proposed fix (Namely, are any modes not recovered in some situations?). I would recommend reproducing the mixture of Gaussians example presented in [1].
A second point is that there are some training-free guidance algorithms that could be included in the comparison to better reflect the state of the art, such as [2] and [3].

[1] Cardoso, G. V., El Idrissi, Y. J., Le Corff, S., & Moulines, E. (2024, January). Monte Carlo guided Diffusion for Bayesian linear inverse problems. In ICLR International Conference on Learning Representations.

[2] Janati, Y., Moufad, B., Qassime, M. A. E., Durmus, A., Moulines, E., & Olsson, J. (2025). A Mixture-Based Framework for Guiding Diffusion Models. arXiv preprint arXiv:2502.03332.

[3] Qu, Q., & Shen, L. (2024). Solving Inverse Problems with Latent Diffusion Models via Hard Data Consistency.

---

> ### Author Rebuttal · Authors · 2025-07-30
>
> We sincerely appreciate your positive feedback, particularly recognizing our contribution as an “original idea to enhance the performance of SMC methods” and for acknowledging the practical relevance of our evaluations using modern generative models. Below, we address the concerns and questions you have raised.
>
> **(1) Additional toy experiment.**
>
> Based on your suggestion, we conducted an additional toy experiment following the Gaussian Mixture (GM) setup described in MCGdiff [1].
>
> For brief summary, the Gaussian mixture setting in MCGdiff [1] defines the data prior $p^{data}$ as a $d$-dimensional Gaussian mixture with 25 components, where the mixture weights are randomly sampled. To simulate a linear inverse problem, a measurement model is introduced with $(y, A, \sigma_y) \in \mathbb{R}^{d_y} \times \mathbb{R}^{d_y \times d} \times [0, 1]$, where $y$ is the observation, $A$ is a linear operator, and $\sigma_y$ is the measurement noise. And the measurement model is randomly sampled. The likelihood is given by $\mathcal{N}(y \mid Ax, \sigma_y^2 I)$ with $x \sim p^{data}$.
>
> We compare the generated samples against those drawn from the analytically tractable posterior $\propto p^{data} \mathcal{N}(y \mid Ax, \sigma_y^2 I)$ using the sliced Wasserstein (SW) distance, where lower values indicate better alignment. We trained a few-step score-based generative model to sample from $p^{data}$.
>
> We evaluate SMC, MALA+SMC, and $\Psi$-Sampler under a fixed computational budget of NFE = 200. We used the same MCMC step size for both MALA+SMC and $\Psi$-Sampler. The sliced Wasserstein (SW) distance is computed for each method, averaged over 20 trials with different random seeds. Experiments are conducted with $d \in \\{80, 800\\}$, while the measurement dimension $d_y$ is varied across $\\{1, 2, 4\\}$.
>
> - $d=80$
> | $d_y$| SMC| MALA+SMC| $\Psi$-Sampler|
> |:----:|:----:|:----:|:----:|
> | 1| 1.9652 ± 0.3285| 1.6669 ± 0.2639| 1.6504 ± 0.2294|
> | 2| 1.3230 ± 0.2755| 1.1045 ± 0.2509| 1.1438 ± 0.2574|
> | 4| 0.4327 ± 0.0741| 0.3447 ± 0.0408| 0.3293 ± 0.0451|
>
> - $d=800$
> | $d_y$| SMC| MALA+SMC| $\Psi$-Sampler|
> |:----:|:----:|:----:|:----:|
> | 1| 2.6379 ± 0.2657| 2.2552 ± 0.2351| 2.1168 ± 0.2275|
> | 2| 1.8778 ± 0.2702| 1.4101 ± 0.2031| 1.3067 ± 0.2285|
> | 4| 1.1826 ± 0.2327| 0.5704 ± 0.0274| 0.5417 ± 0.0400|
>
> The above results show that both posterior-initialization methods (MALA+SMC and $\Psi$-Sampler) outperform standard SMC, which uses particles initialized from a Gaussian prior, even under the same NFE constraint. This indicates that the quality of initial particles influences the performance of SMC, leading to overall improved performance. Notably, our method further outperforms MALA+SMC.
>
> Moreover, the performance gap between MALA+SMC and $\Psi$-Sampler becomes more evident in higher dimensions (i.e., $d=800$) compared to lower ones (i.e., $d=80$). This highlights the effectiveness of pCNL in high-dimensional settings and suggests its importance for extremely high-dimension scenarios such as our main experiment, which operates in a 65,536 dimensional space.
>
> We sincerely thank the reviewer for suggesting this insightful toy experiment. We plan to include a more systematic evaluation on a broader range of toy settings, including this one, in the revision.
>
> **(2) Additional baselines comparison.**
>
> As you pointed out, we reviewed relevant training-free guidance methods [2,3] and conducted additional experiments based on their algorithms. Below are the results on the three applications featured in our main paper, using the same evaluation metrics for consistency. For ease of comparison, we also include the results of DPS [4] and $\Psi$-Sampler as reported in the main paper.
>
> - Layout-to-Image generation
> |  | GroundingDINO [5] (↑) | mIoU [6] (↑)| ImageReward [7] (↑) | VQA [8] (↑) |
> |:----:|:----:|:----:|:----:|:----:|
> | MGDM [2] | 0.204| 0.250| 0.960| 0.720|
> | ReSample [3] | 0.168 | 0.233 | 0.854 | 0.709 |
> | DPS [4] | 0.166| 0.215| 0.705| 0.684|
> | $\Psi$-Sampler| 0.467 | 0.471| 1.035| 0.810 |
>
> - Quantity aware image generation
> |  | T2I-Count [9] (↓) | MAE [10] (↓)| Acc(%) [10] (↑) | ImageReward [7] (↑) | VQA [8] (↑) |
> |:----:|:----:|:----:|:----:|:----:|:----:|
> | MGDM [2] | 12.842 | 14.250 | 2.5 | 0.381|0.933|
> | ReSample [3] | 12.639| 15.175| 0.0| 0.827|0.961|
> | DPS [4]| 14.187| 15.7| 0.0| 0.746 |0.957|
> | $\Psi$-Sampler| 0.850| 2.925| 32.5| 0.796 | 0.951|
>
> - Aesthetic preference image generation
> |  | Aesthetic [11] (↑) | ImageReward [7] (↑) | VQA [8] (↑) |
> |:----:|:----:|:----:|:----:|
> | MGDM [2] | 6.488| 1.078| 0.968|
> | ReSample [3] | 6.193 | 1.179 | 0.957 |
> | DPS [4]| 6.139| 1.116| 0.968|
> | $\Psi$-Sampler| 7.012| 1.171 | 0.963 |
>
> As can be seen from the results, MGDM and ReSample outperform DPS as they are more tailored methods. However, due to the inherent limitations of single-particle methods, their performance still falls short compared to SMC-based methods such as $\Psi$-Sampler. As suggested, we will conduct additional experiments with other training-free guidance methods to further strengthen our analysis and include the results in the revision.
>
> **(3) Clarification on toy experiment.**
>
> We would like to clarify the behavior of MALA+SMC in the toy experiment. The observed inferior performance and its tendency to get stuck in local minima were due to our deliberate choice of using a smaller step size for MALA compared to pCNL. The main motivation behind this decision was to better reflect the actual experimental settings used in the main paper, where MALA requires a smaller step size to avoid extremely low acceptance rates (see Figure 3 in our main paper).
> Indeed, if we were to use the same step size as pCNL in the toy setup, MALA+SMC would also avoid local minima and achieve performance comparable to $\Psi$-Sampler. However, we intentionally matched the toy experiment’s configuration to our main experiments to ensure consistency in illustrating the practical behavior of each sampler.
>
> While MALA+SMC and $\Psi$-Sampler yield similar results in low-dimensional settings—such as the toy experiment in the main paper—when their step sizes are matched, **(1) Additional toy experiment.** demonstrates that $\Psi$-Sampler outperforms MALA+SMC in higher dimensions $d = 800$, even with identical step sizes. This suggests that pCNL offers a distinct advantage in high-dimensional regimes. In extremely high-dimensional cases like our main experiment, MALA suffers from a significantly lower acceptance rate, which necessitates using much smaller step sizes. Attempting to match the step size with pCNL leads to even worse performance for MALA. Therefore, regardless of whether MALA uses a smaller or matched step size, it consistently underperforms compared to $\Psi$-Sampler, resulting in a clear performance gap in the main experiment.
>
> We acknowledge that this point is unclear in the current version of the paper and will revise the text to better clarify the motivation and ensure there is no confusion regarding the toy experiment setup.
>
> **(4) Clarification on the use of Tweedie’s formula approximation.**
>
> We appreciate your suggestion. As you noted, we will revise the discussion around Eq. (11) to explicitly emphasize that the target distribution used for initialization is an approximation derived from Tweedie’s formula.
>
> **(5) Clarification on mode coverage in toy experiment.**
>
> Our claim that $\Psi$-Sampler exhibits “better posterior coverage” was intended to highlight that MALA+SMC samples in the toy experiment tend to deviate from the modes of the ground truth posterior distribution (blue dots in Figure 1(B) in our main paper). As discussed in **(3) Clarification on toy experiment**, we will carefully revise the content related to both the step size and the wording regarding posterior coverage to avoid potential misunderstandings.
>
> **(6) Specification of toy experiment.**
>
> We provide detailed settings for the toy experiment used in the main paper. Specifically, we used a 2D Gaussian mixture with covariance $0.3I$, consisting of six equally weighted components uniformly arranged on a circle of radius 6. The reward function is defined as the sum of three Gaussians centered at points evenly spaced on the circle: $r(x) = e^{-0.5\|x - p_1\|^2} + e^{-0.5\|x - p_2\|^2} + e^{-0.5\|x - p_3\|^2}$, where $p_1=(6.5,0)$, $p_2 = (-3.25,3.25\sqrt{3})$, and $p_3 = (-3.25,-3.25\sqrt{3})$. We trained a few-step score-based generative model using 4-layer MLP with hidden dimension 128. For sampling, we fixed the total NFE to 100 across all methods and visualized the results using 2,000 generated samples. We used step size 0.1 for MALA and 0.2 for pCNL to reflect the step size difference in the main experiment. We will include these details in the appendix for our revised version to ensure reproducibility.
>
> [1] Cardoso et al., Monte Carlo guided Diffusion for Bayesian linear inverse problems, ICLR 2024\
> [2] Janati et al., A Mixture-Based Framework for Guiding Diffusion Models, ICML 2025\
> [3] Song et al., Solving Inverse Problems with Latent Diffusion Models via Hard Data Consistency, ICLR 2024\
> [4] Chung et al., Diffusion posterior sampling for general noisy inverse problems, ICLR 2023\
> [5] Liu et al., GroundingDINO: Marrying DINO with grounded pre-training for open-set object detection, ECCV 2024\
> [6] Hou et al., Salience DETR: Enhancing Detection Transformer with Hierarchical Salience Filtering Refinement, CVPR 2024\
> [7] Xu et al., ImageReward: learning and evaluating human preferences for text-to-image generation, NeurIPS 2023\
> [8] Lin et al., Evaluating text-to-visual generation with image-to-text generation, ECCV 2024\
> [9] Qian et al., T2ICount: Enhancing cross-modal understanding for zero-shot counting, CVPR 2025\
> [10] Amini-Naieni et al., CountGD: Multi-Modal Open-World Counting, NeurIPS 2024\
> [11] Schuhmann, Laion aesthetic predictor, 2022

---

> > ### Comment · Reviewer_bsVN · 2025-08-04
> > **Feedback**
> >
> > I'd like to thank the authors for their rebuttal. I'm satisfied with the answers and with what the authors proposed to add to the paper. I will increase my rating accordingly.

---

> > > ### Author Response · Authors · 2025-08-05
> > >
> > > We are grateful for your positive assessment and for taking the time to review our submission. As mentioned, we will include a systematic evaluation on toy experiments, additional comparisons with a broader range of baselines, as well as the clarifications and specifications discussed.

---

### Official Review · Reviewer_TP7L · 2025-07-02

**Clarity:** 4
**Significance:** 3
**Originality:** 3
**Rating:** 5
**Confidence:** 3

**Summary:**

The paper explores Sequential Monte Carlo (SMC) for inference-time reward alignment, which becomes popular since it does not require fine-tuning. However, existing methods assume a Gaussian prior for initialization. This ignores two issues: (1) the reward signal tends to weaken in later steps, reducing the effectiveness of SMC and (2) modern diffusion models often benefit from better initial points due to mode connectivity in early steps and trajectory straightening.

To address this, the authors propose using pCNL to initialize samples with reward information. Since pCNL keeps the Gaussian prior even in high dimensions, it is suitable for image generation tasks that require high-dimensional noise sampling. The paper compares standard SMC (prior-based), MALA+SMC (posterior-based), and the proposed psi-sampler (pCNL + SMC), and shows that the proposed method performs better.

**Questions:**

- How much does inference time differ between the proposed method and the baselines?
- Could you add one more simple toy example (like Figure 1)? While the three tasks are convincing, such intuitive toy examples would help.
- The method seems closely related to Uehara et al. [20]. Is the main difference that their method requires training, while yours works as inference-time initialization? Could you elaborate more on this difference?

**Ethical Concerns:**

["NO or VERY MINOR ethics concerns only"]

**Final Justification:**

The paper is well-written and demonstrates a thorough review of prior work. It also presents a practically useful technique. The rebuttal also solved the raised questions and concerns such as time comparison, additional data and comparison with the previous work. Therefore, the reviewer raised the final score to accept.

**Limitations:**

yes

**Quality:**

3

**Strengths And Weaknesses:**

Strength
- The motivation is clear and well-explained with literatures of the recent works. (such as SMC-based methods and Uehara et al.)
- It is practically useful since the method improves performance at inference time without extra training.
- The paper shows that pCNL is more effective than other sampling methods such as Top-K-of-N or MALA.
- The method works well across three different practical tasks: layout-to-image, quantity preference, and aesthetic preference.

Weakness
- The main concern is the trade-off between sampling cost and the quality improvement. Since each sample requires a reward query on the predicted x0, the cost may be high.
- It is a bit surprising that the simple Top-K-of-N approach achieves comparable results in some cases, which may reduce the overall impact of the proposed method.
- As the authors mention, the method depends on a reliable reward model.

---

> ### Author Rebuttal · Authors · 2025-07-30
>
> We greatly appreciate your review, especially your recognition of the clear motivation, practical utility of inference-time alignment without additional training, and the effectiveness of our proposed method across multiple tasks. Below, we address the points raised in your review.
>
> **(1) Trade-off between sampling cost and the quality improvement.**
>
> It is true that there is a trade-off between cost and quality. Inference-time scaling refers to enhancing performance by allocating more computational resources during inference rather than training. This enables users to adjust the computational budget at inference time to improve reward alignment, offering greater flexibility and scalability in practical settings. To explicitly demonstrate this trade-off, we conducted an additional experiment where we increased the computational cost and measured the resulting quality. This experiment was conducted using our method on the layout-to-image generation application, with the same evaluation metrics as those used in the main paper.
>
> | NFE | 1000 | 1500 | 2000 | 3000 |
> |:---:|:---:|:---:|:---:|:---:|
> | **GroundingDINO [1] (↑)**   | 0.467  | 0.486  | 0.495  |0.527|
> | **mIoU [2] (↑)**  | 0.471  | 0.468  | 0.476  |0.519|
> | **ImageReward [3] (↑)**  | 1.035  | 1.076  | 1.127  |1.124|
> | **VQA [4] (↑)**  | 0.810  | 0.808  | 0.855  | 0.861 |
>
> As shown in the table above, we observe that the metric improves as the computational budget increases.
>
> **(2) Performance of Top-K-of-N approach.**
>
> Across all applications, our method consistently outperforms Top-K-of-N when evaluated with known rewards. More importantly, even when assessed with task-specific held-out reward metrics that best reflect each application's core objective, our method shows superior performance.
>
> For example, in the layout-to-image generation task, we evaluate using mean Intersection-over-Union (mIoU) measured with an alternative object detector (Salience-DETR [2]), and in quantity-aware image generation task, we use a different counting model (Count-GD [5]) to report MSE and Accuracy (%). In both cases, which serve as strong held-out evaluations, our method surpasses Top-K-of-N.
>
> While the performance is in some case comparable on general-purpose held-out reward metrics like VQA [4] and ImageReward [3], these evaluate image quality and text-prompt alignment more broadly. In contrast, the task-specific held-out metrics more directly reflect the ability to meet the unique conditions of each application.
> Thus, the consistent gains on these application-relevant held-out metrics strongly demonstrate the effectiveness of our approach in task-specific alignment and control.
>
> Furthermore, we would like to emphasize that the Top-K-of-N approach is part of our contribution whioch serve as a novel framework that instantiates our core idea: sampling initial particles for SMC from the reward-informed posterior rather than from the prior. The essence of our contribution lies in demonstrating the importance of posterior-based initialization, and Top-K-of-N offers a simple yet effective approximation of this principle. That even such a rough approximation leads to improved performance strongly supports our central claim.
>
> **(3) Time comparison between proposed method and the baselines.**
>
> We measured wall-clock time using the aesthetic-preference image generation application as an example to compare between different methods. All numbers in the table are reported in seconds, measured using an NVIDIA A100 GPU.
>
> | DPS [6]| FreeDoM [7]| TDS [8]| DAS [9] | Top-K-of-N | ULA+SMC | MALA+SMC | $\Psi$-Sampler|
> |:----:|:----:|:----:|:----:|:----:|:----:|:----:|:----:|
> | 25 | 41| 242 | 240 | 235 | 238| 243 | 241 |
>
> To summarize, all SMC-based methods, including TDS, DAS, Top-K-of-N, ULA+SMC, MALA+SMC, and $\Psi$-Sampler, took approximately 4 minutes, while DPS took 25 seconds and FreeDoM took 41 seconds.
>
> **(4) Additional toy experiment.**
>
> Due to the rebuttal policy, we cannot include any image so we cannot visualize it and thus report quantitative metric for the additional toy experiment. We closely followed the Gaussian Mixture (GM) toy experiment setup described in MCGdiff [10].
>
> Briefly, the experiment considers a $d$-dimensional Gaussian mixture model with 25 components, which serves as the data prior $p^{data}$. The mixture component weights are randomly sampled. To emulate a linear inverse problem, the setting includes a measurement model defined by $(y,A,\sigma_y)\in \mathbb{R}^{d_y}\times \mathbb{R}^{d_y\times d}\times [0,1]$, where $y$ is the measurement, $A$ is a linear operator, and $\sigma_y$ is measurement noise. And the measurement model is randomly sampled. The likelihood is given by $\mathcal{N}(y|Ax, \sigma^2_y I), x\sim p^{data}$.
>
> MCGdiff [10] then compare generated samples with samples drawn from the analytically tractable posterior $\propto p^{data}\mathcal{N}(y|Ax, \sigma^2_y I)$ by computing the sliced Wasserstein (SW) distance (lower values indicate closer alignment between the two distributions ). We trained a few-step score-based generative model to sample from $p^{data}$.
>
> We evaluate SMC, MALA+SMC, and $\Psi$-Sampler under a fixed computational budget of NFE = 200. We used the same MCMC step size for both MALA+SMC and $\Psi$-Sampler. The SW distance is computed for each method, and we report the mean and standard deviation over 20 runs with different random seeds. We conducted experiments with $d\in \\{80, 800\\}$, varying the measurement dimension $d_y \in \\{1, 2, 4\\}$.
>
> - $d=80$
> | $d_y$| SMC| MALA+SMC| $\Psi$-Sampler|
> |:----:|:----:|:----:|:----:|
> | 1| 1.9652 ± 0.3285| 1.6669 ± 0.2639| 1.6504 ± 0.2294|
> | 2| 1.3230 ± 0.2755| 1.1045 ± 0.2509| 1.1438 ± 0.2574|
> | 4| 0.4327 ± 0.0741| 0.3447 ± 0.0408| 0.3293 ± 0.0451|
>
> - $d=800$
> | $d_y$| SMC| MALA+SMC| $\Psi$-Sampler|
> |:----:|:----:|:----:|:----:|
> | 1| 2.6379 ± 0.2657| 2.2552 ± 0.2351| 2.1168 ± 0.2275|
> | 2| 1.8778 ± 0.2702| 1.4101 ± 0.2031| 1.3067 ± 0.2285|
> | 4| 1.1826 ± 0.2327| 0.5704 ± 0.0274| 0.5417 ± 0.0400|
>
> These results demonstrate that both posterior-based initialization methods (MALA+SMC and $\Psi$-Sampler) consistently outperform standard SMC, which initializes particles from a Gaussian prior, even under the same NFE constraint. This highlights the importance of high-quality initial particles in improving SMC performance and achieving overall gains. Among the two, our method achieves better results than MALA+SMC.
>
> Additionally, the advantage of our method over MALA+SMC becomes evident as the dimensionality grows (e.g., $d=800$ vs. $d=80$), illustrating the strength of pCNL in high-dimensional regimes. This further supports the use of pCNL in extremely high-dimension settings like our main experiment, which involves a 65,536-dimensional latent space.
>
> **(5) Difference between Uehara et al. [11] and ours.**
>
> As you correctly pointed out, the main difference is that our method is completely training-free, whereas Uehara et al. [11] does require training. In fact, their method involves training several components. To be more specific, they fine-tune the generative model using a neural SDE solver. In terms of initialization, while we adopt an MCMC method to directly sample from a reward-informed posterior distribution (an approximation of the optimal initial distribution using Tweedie’s formula), they instead train an additional SDE-based model to approximately sample from the optimal initial distribution using neural SDE solver. This also requires learning a value function at time $t=1$, which incurs additional cost for data collection and training. These steps comprise a three-stage training process.
>
> Such a pipeline can be costly and restrictive, especially when applying the method to a new generative model or a different reward function, as all components must be retrained. In contrast, our approach operates entirely at inference time and can be readily applied to new models or rewards without any additional training.
>
> [1] Liu et al., GroundingDINO: Marrying DINO with grounded pre-training for open-set object detection, ECCV 2024\
> [2] Hou et al., Salience DETR: Enhancing Detection Transformer with Hierarchical Salience Filtering Refinement, CVPR 2024\
> [3] Xu et al., ImageReward: learning and evaluating human preferences for text-to-image generation, NeurIPS 2023\
> [4] Lin et al., Evaluating text-to-visual generation with image-to-text generation, ECCV 2024\
> [5] Amini-Naieni et al., CountGD: Multi-Modal Open-World Counting, NeurIPS 2024\
> [6] Chung et al., Diffusion posterior sampling for general noisy inverse problems, ICLR 2023\
> [7] Yu et al., FreeDoM:Training-free energy-guided conditional diffusion model, ICCV 2023\
> [8] Wu et al., Practical and asymptotically exact conditional sampling in diffusion models, NeurIPS 2023\
> [9] Kim et al., Test-time alignment of diffusion models without reward over-optimization, ICLR 2025\
> [10] Cardoso et al., Monte Carlo guided Diffusion for Bayesian linear inverse problems, ICLR 2024\
> [11] Uehara et al., Fine-Tuning of Continuous-Time Diffusion Models as Entropy-Regularized Control, arXiv

---

> ### Comment · Reviewer_TP7L · 2025-08-06
>
> Thank you for the detailed author rebuttal, which helped the reviewer’s understanding. Most of the reviewer’s concerns and questions have been addressed, especially with the additional toy examples demonstrating that the method works well in general. Therefore, the reviewer has raised the final score to accept.

---

> > ### Author Response · Authors · 2025-08-06
> >
> > We truly appreciate your positive assessment of our work and the time you took to review our submission. Your thoughtful feedback is deeply appreciated.

---

### Official Review · Reviewer_UMRC · 2025-07-02

**Clarity:** 3
**Significance:** 3
**Originality:** 3
**Rating:** 4
**Confidence:** 4

**Summary:**

The paper proposes the $\Psi$-sampler, a SMC-based framework for accurately sampling the posterior distribution formed by a pre-trained diffusion model and a reward function. The algorithm contains two components: a pCNL-based MCMC sampling for the initial, non-Gaussian prior distribution, and an SMC-based sampling scheme. $\Psi$-sampler is numerically evaluated on various image generation tasks and demonstrates convincing performance.

**Questions:**

I have the following questions that I would appreciate your answers to. Answers to them are very important for me to maintain a positive support of this paper. Clarifications are certainly welcome if I have some misunderstandings.

1. Related to my major concern: how different is the initial distribution used in $\Psi$ sampler from pure Gaussian, and can you visualize them in some toy examples?

2. Related to my major concern: how would the algorithm perform if we just start from a Gaussian prior and do the SMC resampling only? Ablations on this part are essential to understanding the core contributions of this work. While I think that doing SMC for reward alignment itself is already a significant contribution to the field, it's even more critical to determine what truly works.

3. Related to my minor concern: how much bias would there be by using Tweedie's formula for approximation instead of learning a real value function? I understand that this is a somewhat standard practice in the literature, but I am curious about how much bias would be mitigated when combining it with SMC. It would be beneficial to have some numerical experiments discussing this, as SMC is known to be an asymptotically exact sampling algorithm.

**Ethical Concerns:**

["NO or VERY MINOR ethics concerns only"]

**Final Justification:**

The paper proposes an empirically effective algorithm, and my technical concern about its motivation is well addressed through the author's response. Based on the good empirical results of the method and the novelty, I maintain my recommendation of weak acceptance.

**Limitations:**

yes, limitations are addressed.

**Quality:**

3

**Strengths And Weaknesses:**

**Strength:**
1. The paper is very readable with a clear motivation and easy-to-follow logic.
2. The proposed algorithm is novel and aims at an important application: post training of diffusion models.
3. The numerical experiments are quite extensive and demonstrate the advantage of the proposed algorithms.

**Weakness:**

**I have a major concern regarding the validity of this idea of sampling from "optimal initial distribution" for technical reasons**. As is mentioned in [1], as well as eq (9) in the appendix, the optimal initial distribution is indeed proportional to $p_1(x_1)\exp(V_1(x_1))$, where $v_t$ is the value function. From the expression of value function in eq (11),  the value function is given by
$V_1(x_1)) = \alpha \log \mathbb{E}[\exp(r(x_0)/\alpha) | x_1]$ where this expectation is taken with respect to the coupling between $x_0$ and $x_1$ produced by the path measure given by pretrained diffusion model. This creates an issue, as $x_0$ and $x_1$ are independent when the forward noising process add enough noise (which is usually the case in practice), making the conditional expectation just a regular expectation. Therefore, $V_1(x_1)$ is just a constant function, and the optimal initial distribution is just the Gaussian prior. This is also discussed in [2].

Therefore, I am concerned about the actual significance of starting from a non-Gaussian prior and whether SMC does all the heavy lifting for the performance of $\Psi$-sampler. This creates additional computation overhead due to the use of MCMC (and one of the main novelties the paper claims is built upon this). This would certainly need more clarification and ablations.

Some other minor issues are listed below:
1. Missing some references in related work. I note that continuous time SMC-based diffusion posterior sampling algorithms have been proposed in [3] for image inverse problems, as well as in [4]. While this does not affect my evaluation of the contribution of this paper as these are concurrent, I think it's important to discuss connections to these works for a more comprehensive overview of the field to further improve readability.

2. The introduction of eq (3), (4) might confuse readers. I think it's not emphasized enough that with eq (3), (4), the method **does not converge to the correct posterior distribution** even asymptotically when SMC has large number of particles. This is in contrast to the finetuning-based approach, which indeed recovers the correct distribution. The paragraph in appendix is much better for setting up the problem, and the approximation is more evident. I suggest moving that into the main text.

**References**

[1] Uehara, Masatoshi, et al. "Fine-tuning of continuous-time diffusion models as entropy-regularized control."

[2] Domingo-Enrich, Carles, et al. "Adjoint matching: Fine-tuning flow and diffusion generative models with memoryless stochastic optimal control."

[3] Chen, Haoxuan, et al. "Solving inverse problems via diffusion-based priors: An approximation-free ensemble sampling approach."

[4] Skreta, Marta, et al. "Feynman-kac correctors in diffusion: Annealing, guidance, and product of experts."

---

> ### Author Rebuttal · Authors · 2025-07-30
>
> We greatly appreciate your review, especially your recognition of the paper’s clarity, the novelty of our method, and the importance of the application. Our detailed responses to your concerns are provided below. Please note that the references you mentioned appears to be missing. If you could provide the relevant citations, we would be happy to review it.
>
> **(1) Validity of the idea of sampling from "optimal initial distribution"**
>
> First of all, thank you for insightful question. However, since no reference was provided in the review, we were unable to verify what [2] refers to. If you could share the reference, we would be happy to review the relevant work and provide a more detailed response.
>
> Regarding your comment, in general, $\mathbf{x}_0$ and $\mathbf{x}_1$ are **not** independent unless a very specific diffusion coefficient is chosen—referred to as a *memoryless noise schedule*; for details, please refer to [1].
>
> Let the stochastic interpolant be defined as $\mathbf{x}_t = a_t \mathbf{x}_0 + b_t \mathbf{x}_1$, which bridges the data distribution $\mathbf{x}_0$ and the prior distribution $\mathbf{x}_1$. The stochastic differential equation (SDE) that shares the same marginal distribution takes the form: $d\mathbf{x}_t=\mathbf{f}(\mathbf{x}_t,t)dt+g(t)d\mathbf{W}$ (see our supplementary material, Section A for further details). For $\mathbf{x}_0$ and $\mathbf{x}_1$ to be independent, the diffusion coefficient **should be** $g(t)=\sqrt{2 b_t \left( \dot{b}_t - \frac{\dot{a}_t}{a_t}b_t\right) }$. This condition defines the memoryless noise schedule [1]. Therefore, unless this specific diffusion coefficient is used, $\mathbf{x}_0$ and $\mathbf{x}_1$ are generally not independent, and the value function is not constant. We will include a clear explanation and its implications in the revision.
>
> A natural question that may arise is whether one could simply use the memoryless noise schedule instead. The issue is that its magnitude over time is quite large, which results in significantly more noise being added at each step. Consequently, this requires many more denoising steps to recover a clean image. For example, in our experiments, using a diffusion coefficient of $g(t) = 1 - (t - 1)^2$ enabled the generation of clean images within approximately 10 steps. In contrast, using the memoryless noise schedule ($g(t) = \sqrt{\frac{2t}{1 - t}}$) required several hundred denoising steps to achieve comparable visual quality.
>
> Beyond this, there are several practical reasons why $\mathbf{x}_0$ and $\mathbf{x}_1$ cannot be independent. Since training time and model capacity are finite, the learned model inevitably captures some correlation between noise and the image. This is often reflected in text-to-image generation, where sampling with specific noise inputs can yield better results for a given prompt [2]. Furthermore, many prior [3,4] works attempt to improve generation by selecting or optimizing noise vectors that better match the conditioning input. These studies provide practical evidence that they are in fact dependent in real-world models.
>
> **(2) Significance of starting from a non-Gaussian prior.**
>
> We wish to bring to your attention that Table 1 of the main paper already covers the comparison related to your comment. Specifically, the baselines TDS [5] and DAS [6] are designed to sample initial particles for the subsequent SMC steps from the prior distribution, i.e., a standard Gaussian. To be more specific, TDS is a method that samples initial particles from prior and perform SMC, while DAS builds on this by incorporating a tempering strategy that scales the reward based on the time step. In contrast, our method employs posterior-based initialization before performing SMC. The empirical results show that this difference in initialization leads to a noticeable performance improvement. This indicates that the improvement is not solely due to SMC itself, but also to the quality of the initial particles.
>
> Note that, to ensure a fair comparison, we carefully matched the NFE across all SMC-based methods. We ensured that our method does not consume more NFE than the baselines.
> We will revise the text to more clearly state that we have conducted such experiments.
>
> **(3) Missing references.**
>
> Thank you for pointing this out. However, the references [3] and [4] mentioned in your review were not included, so we were unable to identify which specific works you are referring to. If you could kindly provide the references, we would be happy to review them in detail. We will make sure to incorporate those works in the revision for a more comprehensive overview of the field.
>
> **(4) Clarification of Eq.(3) and Eq.(4).**
>
> Due to space constraints, we were unable to include all related details in the main paper and thus placed them in the supplementary. We agree that this point could be made clearer, and we will revise the main text to better highlight this in the next version.
>
> **(5) Comparing posterior distributions to Gaussian with toy examples.**
>
> Considering the high dimensionality used in our experiments, it is challenging to provide an intuitive illustration of the difference between the prior and posterior distributions. This is precisely why we included the toy experiment shown in Figure 1 of the main paper. In Figure 1(B), the blue dots represent samples from the posterior distribution, which exhibit a clear deviation from the Gaussian prior illustrated as blue dots in Figure 1(A).
>
> **(6) Ablation on initialization strategies for SMC.**
>
> As mentioned in **(2) Significance of starting from a non-Gaussian prior.**, both TDS [5] and DAS [6] initialize particles from a Gaussian prior and apply SMC. These baselines are included in Table 1 of the main paper, and the results show that the method that samples initial particles from the posterior (labeled as **Sampling from Posterior** in Table 1) achieves better performance under the same computational budget. To clarify this point, we will revise the text accordingly.
>
> **(7) Bias of Tweedie's formula approximation.**
>
> First of all, regarding the mitigation of bias arising from the Tweedie approximation using SMC, the two are not directly related. The bias stems from the approximation itself where the posterior is approximated by a Dirac-delta distribution centered at the posterior mean, and SMC cannot eliminate this bias.
>
> Next, regarding the extent of bias introduced by Tweedie’s formula approximation, the resulting bias depends on how much the true posterior deviates from this point-mass approximation. When the posterior is sharp, the approximation becomes more accurate.
> Current few-step score-based generative models typically shows strong correlation between $\mathbf{x}_t$ and $\mathbf{x}_0$, leading to a sharper posterior and thus a better approximation.
>
> In contrast, learning a value function introduces significant additional complexity, such as the need for reward data collection, architectural choices, and stability challenges during training. Moreover, value function training is highly sensitive to reward scale, distribution shift, and task-specific tuning, which can make generalization to new tasks difficult.
> Tweedie’s formula approximation, on the other hand, offers a simple yet effective alternative that is easy to compute, requires no additional training, and generalizes readily across tasks.
>
> [1] Dormingo-Enrich et al., Adjoint Matching: Fine-tuning Flow and Diffusion Generative Models with Memoryless Stochastic Optimal Control, ICLR 2025\
> [2] Zhou et al., Golden Noise for Diffusion Models: A Learning Framework, ICCV 2025\
> [3] Guo et al., INITNO: Boosting Text-to-Image Diffusion Models via Initial Noise Optimization, CVPR 2024\
> [4] Wang et al., The Silent Assistant: NoiseQuery as Implicit Guidance for Goal-Driven Image Generation, ICCV 2025\
> [5] Wu et al., Practical and asymptotically exact conditional sampling in diffusion models, NeurIPS 2023\
> [6] Kim et al., Test-time alignment of diffusion models without reward over-optimization, ICLR 2025

---

> > ### Comment · Reviewer_UMRC · 2025-08-02
> >
> > Thank you for the response! Sorry for forgetting to list the references [1-4] in the original review. They are listed below.
> >
> > The concerns raised due to [1-2] are already well addressed, and I appreciate the explanation. For [3-4], I think it would be essential to incorporate them in the literature review, especially [3], which also works on the image posterior sampling problem using SMC. Overall, most concerns are well taken care of, and thus, I am happy to maintain a positive rating about this paper, given that the promised revision is implemented. Good luck!
> >
> > **References**
> >
> > [1] Uehara, Masatoshi, et al. "Fine-tuning of continuous-time diffusion models as entropy-regularized control."
> >
> > [2] Domingo-Enrich, Carles, et al. "Adjoint matching: Fine-tuning flow and diffusion generative models with memoryless stochastic optimal control."
> >
> > [3] Chen, Haoxuan, et al. "Solving inverse problems via diffusion-based priors: An approximation-free ensemble sampling approach."
> >
> > [4] Skreta, Marta, et al. "Feynman-kac correctors in diffusion: Annealing, guidance, and product of experts."

---

> > > ### Author Response · Authors · 2025-08-03
> > >
> > > We sincerely appreciate your time and effort in reviewing our submission and rebuttal, and thank you for providing the missing references. Regarding [3] and [4], we will incorporate both works into the related work section and include a detailed discussion of their relevance and distinctions from our approach.

---

### Official Review · Reviewer_P4YQ · 2025-07-03

**Clarity:** 3
**Significance:** 4
**Originality:** 4
**Rating:** 5
**Confidence:** 3

**Summary:**

This paper investigates the limitations of the existing SMC framework, i.e. the initial particles have to be coming from the standard Gaussian prior. This limitation will give very poor coverage over the high-probability region. Also, it can significantly impeded the sampling efficiency. Therefore, this paper proposes a new approach to put the initial particles in more likely regions through pCNL. This reward-informed initialization ensures that the particle set is better aligned with the target distribution from the outset, which will make the sampling more efficient and accurate.
The paper shows a crude approximation of Top-K-of-N method, which works surprisingly well. Therefore, the paper adopts this crude and simple approach to do SMC sampling on layout-to-image generation tasks. Empirical results show that the model is able to outperform the baseline posterior-based sampling methods like ULA, MALA, etc significantly.

**Questions:**

N/A

**Ethical Concerns:**

["NO or VERY MINOR ethics concerns only"]

**Final Justification:**

I have read the rebuttal and decide to maintain my score.

**Limitations:**

The paper is well written. The main limitation is that the bridge between the theorem and the application. The authors have not stated clearly why they chose the three specific evaluation tasks.

**Quality:**

4

**Strengths And Weaknesses:**

Strength:
1. The paper proposes a very solid approach, which has a significant technical contribution.
2. The paper's theorem is highly sound.
3. The paper's results also reflect the advantages of the proposed method.
4. The paper is written quite well. The motivation and background has been written in details to help the readers understand the limitations of the existing methods.

Weakness:
1. It's not particularly clear why the three specific applications are chosen. Why don't the authors use widely adopted benchmarks like COCO-image generation or Flickr-image generation? Is there a reason to not do normal text-to-image generation evaluation and focus on the more specific ones?

---

> ### Author Rebuttal · Authors · 2025-07-28
>
> We greatly appreciate your thoughtful review, especially your recognition of the paper’s strong technical contribution and clear writing. Below, we address the specific points raised.
>
> **(1) Text-to-image generation application.**
>
> We conducted text-to-image generation experiment using the COCO-2014 dataset. Specifically, we randomly sampled 100 prompts and employed PickScore [1] as the reward model (which measures prompt alignment of the image). To evaluate generalization, we used VQAScore [2] as a held-out reward model (an evaluation metric that is not accessible during generation) which is also used in the main paper to assess prompt alignment. For score-based generative model, we used FLUX-Schnell [3] as in main experiment in the paper.
>
> |   | DPS [4] | FreeDoM [5] | TDS [6] | DAS [7] | Top-K-of-N | ULA+SMC| MALA+SMC | $\Psi$-Sampler|
> |:------:|:------:|:------:|:------:|:------:|:------:|:------:|:------:|:-------:|
> | PickScore [1] (↑) | 0.224 | 0.226| 0.239| 0.239 | 0.239 | 0.238| 0.239| 0.240 |
> | VQA [2] (↑) | 0.864| 0.897 | 0.908 | 0.913 | 0.915| 0.893| 0.911 | 0.918 |
>
> As shown in the table above, SMC-based methods (TDS, DAS, Top-K-of-N, ULA+SMC, MALA+SMC, $\Psi$-Sampler) outperform single-particle methods such as DPS and FreeDoM. However, the differences among the SMC-based methods themselves are minor. The reason is that text-to-image generation has become relatively easy for current large-scale text-to-image generation models. As a result, when applying reward-guided sampling to this task, performance quickly saturates regardless of the specific method used, making it difficult to observe meaningful differences between approaches.
>
> Therefore, in the main experiment, we used alternative applications that are both practically relevant and more challenging. Aesthetic preference image generation is one such example, as it is a commonly used application in several recent works [7,8,9]. To evaluate our approach under more challenging reward conditions, we selected layout-to-image and quantity-aware image generation tasks, which are significantly harder for conventional text-to-image models to handle because they require precise control and fine-grained alignment with the input specification.
>
> In addition to the three applications presented, we plan to enrich the paper with a broader set of experiments to explore the versatility and potential of our approach. Specifically, we aim to include additional applications such as style-guided image generation and segmentation map-guided image generation, which would offer a more comprehensive analysis and demonstrate the broader applicability of our method.
>
> [1] Kirstain et al., Pick-a-pic: An open dataset of user preferences for text-to-image generation, NeurIPS 2023\
> [2] Lin et al., Evaluating text-to-visual generation with image-to-text generation, ECCV 2024\
> [3] Black Forest Labs, FLUX, 2024\
> [4] Chung et al., Diffusion posterior sampling for general noisy inverse problems, ICLR 2023\
> [5] Yu et al., FreeDoM:Training-free energy-guided conditional diffusion model, ICCV 2023\
> [6] Wu et al., Practical and asymptotically exact conditional sampling in diffusion models, NeurIPS 2023\
> [7] Kim et al., Test-time alignment of diffusion models without reward over-optimization, ICLR 2025\
> [8] Uehara et al., Bridging Model-Based Optimization and Generative Modeling via Conservative Fine-Tuning of Diffusion Models, NeurIPS 2024\
> [9] Black et al., Training Diffusion Models with Reinforcement Learning, ICLR 2024

---

### Note · Authors · 2025-08-14

Dear AC and Reviewers,

We are encouraged that **all reviewers gave initially positive scores and further converged toward acceptance through the rebuttal and discussion phase.** We sincerely thank the Area Chair and reviewers for their time and effort in reviewing our submission. We greatly appreciate the constructive and thoughtful feedback.

Reviewers also recognized several key strengths of our work. They highlighted the novelty and practicality of our inference-time reward alignment approach, describing it as “an original idea to enhance considerably the performance of SMC methods” (bsVN) and “a novel algorithm targeting an important application” (UMRC). The paper was also praised for its clarity and motivation (UMRC, P4YQ), and the empirical results were seen as “extensive and convincing,” demonstrating effectiveness “across three practical tasks” (TP7L).

We are also pleased to have addressed all major concerns during the rebuttal. For example, we added new toy experiments following the MCG-Diff setup (suggested by bsVN), showing that our method works well in general. We also extended comparisons with recent baselines such as MGDM and ReSample, and demonstrated superior performance. Further, we clarified theoretical points—such as why the value function is not constant in general (UMRC).

Before wrapping up, we want to emphasize our key technical contributions as follows:
- **Posterior-based initialization matters.** \
We identify the problem and limitations of prior-based initialization and demonstrate that sampling SMC particles from a reward-informed posterior is crucial for improved performance. Even a simple instantiation such as Top-K-of-N shows clear gains, supporting our central claim.
- **High-dimensional, robust sampling via pCNL.** \
Among feasible MCMC choices, we advocate the use of pCNL for constructing initial particles. We show that pCNL offers stable acceptance rates and effective mixing in the extremely high-dimensional latent spaces of modern score-based generative models, leading to overall performance gain.
- **Broad improvements.** \
Under a fixed NFE budget, $\Psi$-Sampler consistently improves reward alignment across tasks including layout-to-image generation, quantity-aware generation, and aesthetic-preference generation, with strong generalization to held-out rewards.

Thank you again for the time and effort for providing valuable comments. We will further revise and enrich the paper based on the reviewers’ suggestions.

---

### Decision · Program_Chairs · 2025-09-17

**Decision:**

Accept (spotlight)

**Comment:**

The paper improves inference-time reward alignment by better initilisation of particles in SMC. The main conceptual contribution is using a reward-aware posterior and the pCNL sampling algorithm. Extensive numerical experiments support the claim of improvement performance compared to SOTA for a number of different tasks.
The paper is well written, cites appropriate literature and present a systematic analysis of the method. During the rebuttal, the reviewers identified some weaknesses related to the experiments, tradeoffs and novelty, and the authors provided convincing responses that address all the main concerns. 3 reviewers proposed a score of 5 and one of 4. I agree on the soundness of the paper and its impact for applications of score based models.